 **eLIFE**

# Step-to-step variations in human running reveal how humans run without falling

**Nidhi Seethapathi[1,2]\*, Manoj Srinivasan[1]\***

[1]Mechanical and Aerospace Engineering, The Ohio State University, Columbus, United States; [2]Department of Bioengineering, University of Pennsylvania, Philadelphia, United States

**Abstract** Humans can run without falling down, usually despite uneven terrain or occasional pushes. Even without such external perturbations, intrinsic sources like sensorimotor noise perturb the running motion incessantly, making each step variable. Here, using simple and generalizable models, we show that even such small step-to-step variability contains considerable information about strategies used to run stably. Deviations in the center of mass motion predict the corrective strategies during the next stance, well in advance of foot touchdown. Horizontal motion is stabilized by total leg impulse modulations, whereas the vertical motion is stabilized by differentially modulating the impulse within stance. We implement these human-derived control strategies on a simple computational biped, showing that it runs stably for hundreds of steps despite incessant noise-like perturbations or larger discrete perturbations. This running controller derived from natural variability echoes behaviors observed in previous animal and robot studies.
DOI: https://doi.org/10.7554/eLife.38371.001

**\*For correspondence:**
snidhi@seas.upenn.edu (NS);
srinivasan.88@osu.edu (MS)

**Competing interests:** The authors declare that no competing interests exist.

## Introduction

Human running is often modeled as being periodic (*Blickhan and Full, 1993*; *Seyfarth et al., 2002*; *Srinivasan and Holmes, 2008*). But running is not exactly periodic, even on a treadmill at constant speed. Body motion during running varies from step to step (*Cavanagh et al., 1977*; *Belli et al., 1995*; *Jordan et al., 2007*; *Jordan and Newell, 2008*). This step-to-step variability could be due to internal perturbative sources like muscle force noise and sensory noise (*Warren et al., 1986*; *Harris and Wolpert, 1998*; *Osborne et al., 2005*) or small external perturbations (e.g. visual field inhomogeneity, small ground imperfections). To run without falling, the body's 'running controller' must prevent the effects of these small perturbations from growing too large. Here, we provide an experimentally derived low-dimensional characterization of this control that reveals how humans run without falling down.

One classic modeling paradigm for running control assumes that the human leg behaves like a linear spring (*Blickhan, 1989*; *McMahon and Cheng, 1990*; *Blickhan and Full, 1993*). This paradigm has been used to argue how passive-elastic properties may reduce muscle work needed for locomotion (*Alexander and Vernon, 1975*; *Alexander, 1990*) and has been useful in examining locomotion in a simplified setting. Variants of these spring-mass running models have demonstrated stable running (*Seyfarth et al., 2002*; *Seipel and Holmes, 2005*; *Ghigliazza et al., 2005*; *Geyer et al., 2006*; *Srinivasan and Holmes, 2008*; *Englsberger et al., 2016*). These models have been successful in fitting the average center of mass motion during running (*Blickhan and Full, 1993*; *Geyer et al., 2006*; *Srinivasan and Holmes, 2008*). However, understanding running stability requires understanding how deviations from the average motion are controlled. It has been previously recognized that spring-like leg mechanics cannot explain how deviations from the average motion are controlled and eventually attenuated (e.g. *Ghigliazza et al., 2005*; *Biewener and Daley, 2007*; *Maus et al.,*

**eLife digest** Running at a constant speed seems like a series of repetitive, identical strides, but it is not. There are small variations in each stride. Self-inflicted errors in the forces generated by the muscles, or misperceptions from the senses, may cause these tiny imperfections. Uneven terrain or other outside forces, like a push, can also cause changes in a running stride. People must correct for these small changes as they run to avoid falling down. The only way to correct errors in a stride is by changing the force exerted on the ground by the leg.

Now, Seethapathi and Srinivasan document step-by-step how people correct for small imperfections in their running stride to avoid falling. In the experiments, eight people ran on a treadmill at three different speeds while the motion of their torso and each foot was measured along with the forces of each foot on the treadmill. Seethapathi and Srinivasan found that these runners corrected for minor deviations by changing where each foot lands and how much force each leg applies to the treadmill. The runners placed their foot at a different position on each step and these varying foot positions could be predicted by the errors in the body movement between steps. These errors in body movement could also be used to predict how the runners would change the forces applied by their legs on each step. Imperfections in the stride were mostly corrected within the next step. Errors that would cause the runner to fall to the side were corrected more quickly than errors in forward or backward motion. Seethapathi and Srinivasan incorporated these corrective strategies into a computer simulation of a runner, resulting in a simulated runner that did not fall even when pushed.

These findings may inform the design of robots that run more like humans. They may also help create robotic exoskeletons, prosthetic legs and other assistive devices that help people with disabilities move more fluidly and avoid falling.

DOI: https://doi.org/10.7554/eLife.38371.002

_2015_). Here, we examine the role of active muscle control in running stability, using more general models of human locomotion rooted in Newtonian mechanics (_Srinivasan, 2011_).

One way of characterizing the running controller is to apply perturbations (for instance, pushes or pulls or sudden changes in terrain) and examine how the body recovers from the perturbations (_Van Woensel and Cavanagh, 1992_; _Daley and Biewener, 2006_; _Qiao and Jindrich, 2014_; _Riddick and Kuo, 2016_). Instead of such external perturbations, here, we use the naturally occurring step-to-step variability (_Hurmuzlu and Basdogan, 1994_; _Maus et al., 2015_) to characterize the controller. Previous attempts at examining such variability for controller information focused only on walking (_Hurmuzlu and Basdogan, 1994_; _Wang and Srinivasan, 2012_; _Wang, 2013_; _Wang and Srinivasan, 2014_) or considered variants of the spring-mass model (_Maus et al., 2015_). Here, we directly characterize the control in terms of how humans modulate their leg force magnitude and direction during running. The only way to control the center of mass motion is for the leg to systematically change the forces and the impulses it applies on the ground. We uncover how such center of mass control is achieved. We then implement this human-derived controller on a simple mathematical model of a biped (_Srinivasan, 2011_), showing that this biped model runs without falling down, despite incessant noise-like perturbations, large external perturbations, and on uneven terrain.

A human-derived controller such as the one proposed here could inform monitoring devices to quantify running stability or fall likelihood (_O'Loughlin et al., 1993_), or could help understand running movement disorders. Further, implementing such controllers into robotic prostheses and exoskeletons (_Dollar and Herr, 2008_; _Shultz et al., 2015_) will allow the human body to interact more 'naturally' with the device, rather than having to compensate for an unnatural controller. Some running robots have demonstrated stable running, using a variety of control schemes (_Raibert, 1986_; _Chevallereau et al., 2005_; _Tajima et al., 2009_; _Nelson et al., 2019_). But these robots fall short of human performance and versatility. Understanding human running may lead to better running robots.

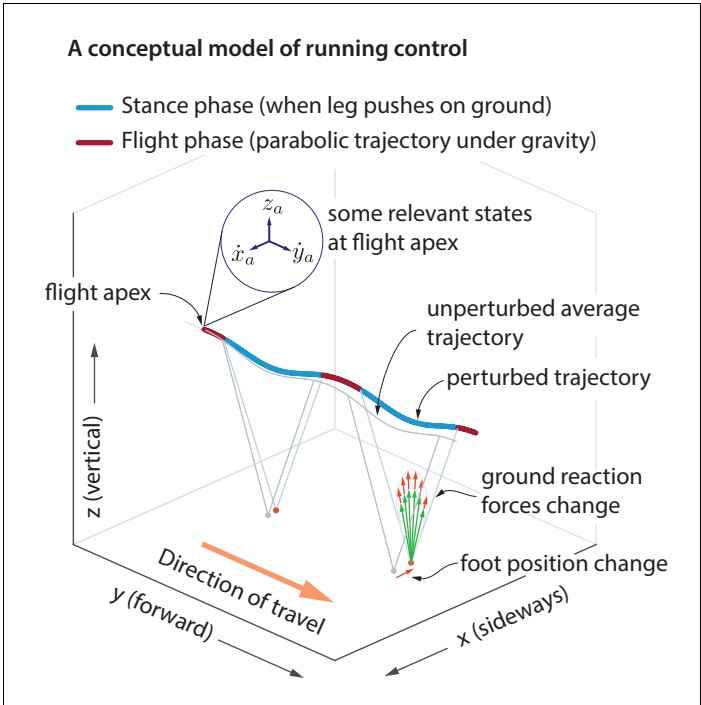

**Figure 1.** Idealized running control. A schematic of a running trajectory, perturbed sideways during flight. The runner can recover back to steady state by changing the ground reaction force — say, by altering the foot placement and the leg force magnitude. This conceptual model is supported by our analysis of running data.

DOI: https://doi.org/10.7554/eLife.38371.004

## Results

The step-to-step variability during running appears superficially random. We show that this variability contains low-dimensional structure, specifically containing information about control strategies involved in running stably. We implement these strategies into a feedback controller, thereby stabilizing a simple mathematical model of a biped, and make further predictions.

We measured body motion and ground reaction forces (GRFs) of humans running for hundreds of steps on a treadmill at three speeds: 2.5, 2.7 and 2.9 m/s (see Materials and methods). Each running step consists of a 'flight phase' with neither foot on the ground, and a 'stance phase' with one foot on the ground. *Figure 1* shows the coordinate system and sign convention: $x$ is sideways, $y$ is fore-aft, and $z$ is vertical. The results we present are for the highest running speed and we discuss speed-dependence of our results separately. Unless otherwise specified, all quantities and results in equations and figures are non-dimensionalized using body mass $m$, acceleration due to gravity $g$, and leg length $\ell_{\max}$. Forces are normalized by $mg$, distances by $\ell_{\max}$, speeds by $\sqrt{g\ell_{\max}}$, time by $\sqrt{\ell_{\max}/g}$, impulses by $m\sqrt{g\ell_{\max}}$, etc.

### A hypothesized controller structure for stable running

During flight phase, the body center of mass moves in a nearly parabolic trajectory and the runner has no control over this parabolic motion (as the aerodynamic forces generated by the person are negligible, unlike birds). From Newton's second law, it follows that the *only way* to control the center of mass motion is to modulate the total ground reaction force components during stance phase, when the leg is in contact with the ground. However, there are infinitely many ways to modulate the ground reaction forces to control the center of mass motion. Here, we examine *how* the ground reaction forces are modulated in response to center of mass state deviations during the previous flight apex (*Figure 1*). A flight apex is defined as when the center of mass height $z$ is maximum. The center of mass position and velocity at flight apex are denoted by $(x_a, y_a, z_a)$ and $(\dot{x}_a, \dot{y}_a, \dot{z}_a)$, respectively. Because the vertical velocity at flight apex $\dot{z}_a = 0$ by definition, $\dot{z}_a$ is not considered as an explanatory variable. The absolute horizontal position $(x_a, y_a)$ on the treadmill changes with a much

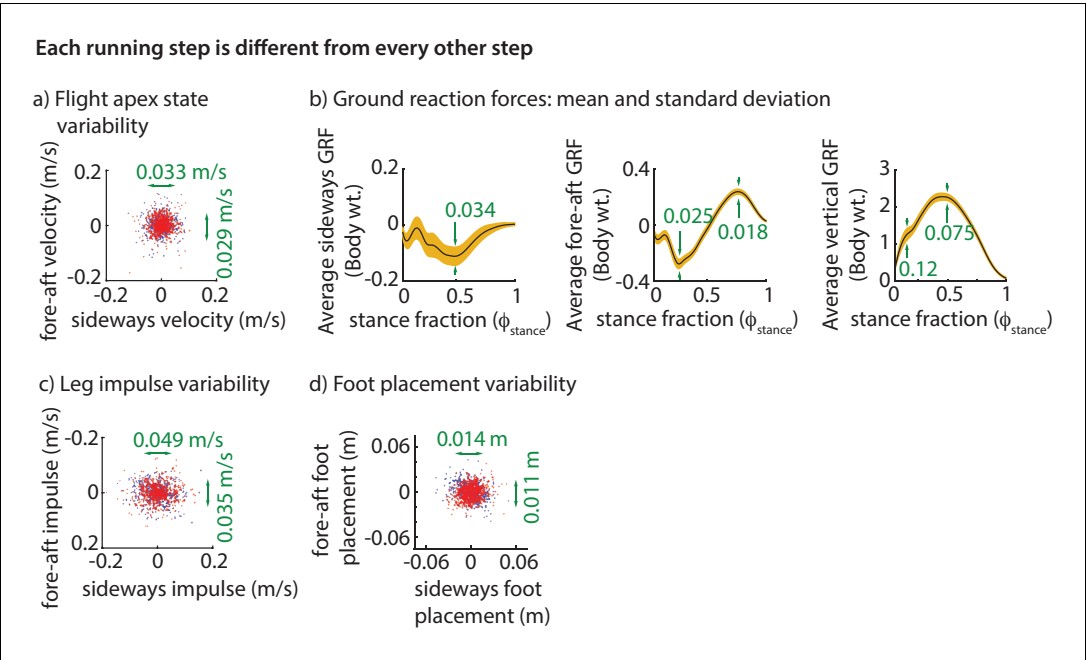

**Figure 2.** Step-to-step variability during running. (**a**) Variability in the center of mass velocity at flight apex. (**b**) Mean GRFs in three directions (black line) and one standard deviation around the mean (yellow band) for a right stance phase; left stance phase GRF is similar in fore-aft and vertical directions, but the sideways GRF is negative of that for the right stance. Green text indicates standard deviation values in all panels. GRFs are in fraction of body weight. (**c**) Variability in the fore-aft and sideways impulses due to the GRFs. (**d**) Variability in foot position relative to torso at the beginning of stance phase. In panels a, c and d each dot corresponds to a separate step and the scatter plot is for all subjects with each subject's mean value subtracted, so that only variability about the mean is shown for 500 randomly chosen steps. Red dots denote right steps and blue dots denote left steps.
DOI: https://doi.org/10.7554/eLife.38371.003

slower time-scale than other variables. Therefore, our default set of explanatory variables is $(\dot{x}_a, \dot{y}_a, z_a)$. We will include the horizontal position $(x_a, y_a)$ when we comment later on 'station keeping'.

## Fore-aft and sideways impulses independently control center of mass motion

The step-to-step variability in the center of mass state at flight apex over hundreds of steps is shown in *Figure 2a*. To be stable, the runner needs to prevent this motion variability from growing without bound. As noted, the only way to control this motion is by using the ground reaction forces (GRFs). Consequently, the ground reaction force components over the stance phase are also variable (*Figure 2b*).

The net effect of the ground reaction forces on the center of mass velocity over a stance phase is captured by the force impulse, namely, the integral of the force. The variability in the sideways and fore-aft ground reaction impulses over a step (*Figure 2c*) are well-predicted by the variability in the center of mass state $(\dot{x}_a, \dot{y}_a, z_a)$ at the previous flight apex (*Figure 3*). Moreover, the sideways impulse depends primarily on the sideways velocity $\dot{x}_a$ and the fore-aft impulse depends primarily on the fore-aft velocity $\dot{y}_a$. Thus, it appears that the control in the fore-aft and sideways directions are independent or decoupled. Pooled over all subjects, the best-fit linear model for the sideways impulse $P_x$ is:

$$\text{Left stance: } \Delta P_x = -1.03\,\Delta\dot{x}_a, \text{ with } R^2 = 0.55, \text{ and}$$
$$\text{Right stance: } \Delta P_x = -1.07\,\Delta\dot{x}_a, \text{ with } R^2 = 0.53, \tag{1}$$

and that for the fore-aft GRF impulse $P_y$ is:

$$\text{Left stance: } \Delta P_y = -0.72\,\Delta \dot{y}_a, \text{ with } R^2 = 0.32, \text{ and}$$
$$\text{Right stance: } \Delta P_y = -0.72\,\Delta \dot{y}_a, \text{ with } R^2 = 0.33,$$
(2)

as in *Figure 3*. All coefficients in *Equations (1) and (2)* are significant at $p < 10^{-4}$. Both sideways and fore-aft impulses depend negligibly on vertical position deviations, so that including $z_a$ in the regression increases the $R^2$ values by less than 0.02.

## Almost deadbeat: impulses correct horizontal velocity mostly within a step

The linear models for the fore-aft and sideways impulses in *Equations (1) and (2)* have a simple interpretation. The $\Delta \dot{x}_a$ coefficient of about $-1$ in *Equation (1)* (that is, $\Delta P_x \approx -\Delta \dot{x}_a$) implies that sideways velocity deviations are completely corrected in one step, on average (over all steps and all subjects). This correction could have been done over many steps, as would be the case if the coefficient were $-0.5$, say. But humans seem to exhibit a 'one-step dead-beat controller' on average for sideways velocity deviations (the term deadbeat control refers to when state deviations decay to zero in a finite amount of time). Of course, this single-step correction is not perfect. An $R^2$ value of about 0.55 suggests that the system over-corrects or under-corrects deviations for any given step.

Analogously, the coefficient of $-0.72$ in *Equation (2)* suggests that about 72% of a forward velocity deviation is corrected in a single step, on average. While this is not strictly 'deadbeat control', it results in only $(1 - 0.72)^2 = 0.08$ of a perturbation remaining after two steps, and $(1 - 0.72)^3 = 0.02$ of a perturbation after three steps, indicating rapid control.

## Apex-to-apex maps also show fast decay of center of mass deviations

We corroborate the above findings regarding perturbation decay with the 'apex-to-apex maps': that is, linear models that describe the relation between deviations in the state at one flight apex and those at the next flight apex. The right-to-left map from the state $S_{\text{right}} = [\dot{x}_a, \dot{y}_a, z_a]_{\text{right}}$ at an apex preceding a right stance to the state at the next flight apex (preceding a left stance) is, approximately:

$$\begin{bmatrix} \dot{x}_a \\ \dot{y}_a \\ z_a \end{bmatrix}_{\text{left}} = K_{\text{R} \to \text{L}} \cdot \begin{bmatrix} \dot{x}_a \\ \dot{y}_a \\ z_a \end{bmatrix}_{\text{right}} \quad \text{where } K_{\text{R} \to \text{L}} = \begin{bmatrix} -0.05^* & -0.02^* & +0.31^* \\ -0.08^* & +0.27 & -0.15^* \\ +0.02^* & +0.06 & +0.46 \end{bmatrix},$$
(3)

where the superscript $^*$ indicates that the coefficient is not significantly different from zero ($p > 0.05$).

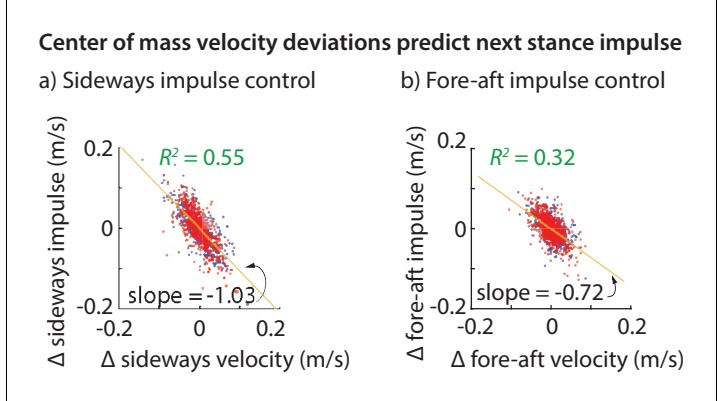

**Center of mass velocity deviations predict next stance impulse**

a) Sideways impulse control    b) Fore-aft impulse control

**Figure 3.** Center of mass velocity deviations predict stance impulse. A linear model based on center of mass velocity deviations at flight apex explains a considerable fraction of the impulse on the next step. The impulses are mass-normalized. The scatter plot shows 500 randomly selected steps for left (blue dots) and right stances (red dots). The best-fit line (yellow) and the corresponding slope are shown for the left stance impulse. The slopes of the best fit line suggest that the sideways impulse corrects about 100% of the sideways velocity deviation on average and the fore-aft impulse corrects about 70% of the fore-aft velocity deviation.
DOI: https://doi.org/10.7554/eLife.38371.005

The left-to-right matrix $K_{\mathrm{L}\rightarrow\mathrm{R}}$ is similar to $K_{\mathrm{R}\rightarrow\mathrm{L}}$, except for the sign changes due to mirror-symmetry. The matrix product of $K_{\mathrm{L}\rightarrow\mathrm{R}}$ and $K_{\mathrm{R}\rightarrow\mathrm{L}}$ — Jacobians of the Poincare map (*Hurmuzlu and Basdogan, 1994*; *Guckenheimer and Holmes, 2013*; *Maus et al., 2015*) — quantify how apex state deviations grow or decay over one stride (two steps). The eigenvalues of this matrix product were all less than one in absolute value, indicating a stable periodic motion. The largest eigenvalue was 0.14, indicating that at most 14% of a perturbation remains after a stride on average in any direction.

The low value of $K_{\mathrm{R}\rightarrow\mathrm{L}}(1,1)$, not significantly different from zero, suggests that a purely sideways velocity perturbation gets corrected essentially over one step on average, consistent with the sideways impulse control (*Equation 1*). Similarly, the value $K_{\mathrm{R}\rightarrow\mathrm{L}}(2,2) = 0.27$ suggests that 73% of a forward velocity deviation is corrected in one step, consistent with the fore-aft impulse control (*Equation 2*). Finally, the $(3,3)$ element of the step map (*Equation 3*) suggests that less than 50% of a deviation in vertical position $z_a$ remains after a step. See (*Maus et al., 2015*) for a detailed Floquet analysis of human running including more state variables, complementing the simplified version here.

## Within-step vertical impulse modulations control vertical position

The control of vertical position is qualitatively different from that of control in the fore-aft and sideways directions, as we cannot use net vertical impulse for vertical position control due to the impulse-momentum considerations below. A flight apex occurs when the center of mass vertical velocity is zero. So, the net vertical impulse between two consecutive flight apexes is also zero (as it equals the change in vertical momentum, according to the impulse-momentum equation). Therefore, changing the net vertical impulse over a stance phase will not accomplish any meaningful control in the vertical direction. However, we will show that by differentially modulating the vertical impulse within one stance phase, we can change the vertical position ($z_a$) from one flight apex to the next, without changing the net impulse.

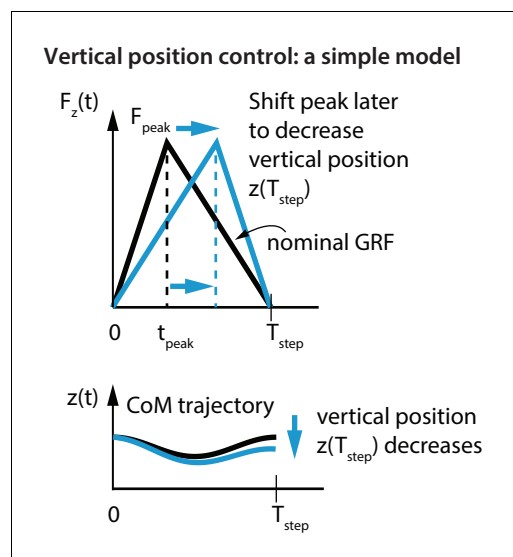

**Figure 4.** Vertical position control by differential impulse control. Using a unimodal vertical GRF, we find that the way to lower vertical position over a step is to move the peak force to the right. This is equivalent to increasing the vertical impulse on the second half of the step and decreasing the vertical impulse over the first half. Conversely, to increase the vertical position over a step, we find that the peak force needs to be moved to the left.

DOI: https://doi.org/10.7554/eLife.38371.006

To show this most simply, consider infinitesimal flight phases and a stance phase from $t = 0$ to $t = T_{\mathrm{step}}$. The total impulse $P_z$ due to the vertical ground reaction force $F_z(t)$ equals that due to gravity, which is given by, $P_z = \int_0^{T_{\mathrm{step}}} F_z(t)\,dt = \int_0^{T_{\mathrm{step}}} mg\,dt = mgT_{\mathrm{step}}$. For a triangular stance force (*Figure 4*) with peak force $F_{\mathrm{peak}}$ at $t_{\mathrm{peak}}$, we get $F_{\mathrm{peak}} = 2mg$. Then, by integrating the vertical acceleration $(F_z/m - g)$ twice, the change in vertical position $z(T_{\mathrm{step}}) - z(0)$ over a step is given by:

$$z(T_{\mathrm{step}}) - z(0) = \frac{g}{6}\left((T_{\mathrm{step}} - t_{\mathrm{peak}})^2 - t_{\mathrm{peak}}^2\right). \quad (4)$$

If the step was symmetric about mid-stance ($t_{\mathrm{peak}} = T_{\mathrm{step}}/2$), there is no vertical position change over a step ($z(T_{\mathrm{step}}) = z(0)$). The flight apex vertical position on the next step $z(T_{\mathrm{step}})$ can be changed by changing $t_{\mathrm{peak}}$ relative to $T_{\mathrm{step}}/2$ (*Figure 4*). For example, if $z(0)$ at one flight phase was greater than its nominal value and the runner wishes to reduce it, this simple model predicts that the runner will decrease the first-half impulse and increase the second-half impulse; doing this is equivalent to delaying $t_{\mathrm{peak}}$ relative to $T_{\mathrm{step}}/2$ (as in *Figure 4*). This prediction is in agreement with the following experimentally-derived linear relations for the first half vertical impulse from $t = 0$ to $T_{\mathrm{step}}/2$, namely $\Delta P_z|_0^{T_{\mathrm{step}}/2}$, and the second half

vertical impulse from $t = T_{\text{step}}/2$ to $T_{\text{step}}$, namely $\Delta P_z|_{T_{\text{step}}/2}^{T_{\text{step}}}$:

$$\text{Left stance: } \Delta P_z|_0^{T_{\text{step}}/2} = -2.5\,\Delta z_a \text{ and } \Delta P_z|_{T_{\text{step}}/2}^{T_{\text{step}}} = +2.5\,\Delta z_a \text{ with } R^2 = 0.35, \text{ and} \tag{5}$$

$$\text{Right stance: } \Delta P_z|_0^{T_{\text{step}}/2} = -2.3\,\Delta z_a \text{ and } \Delta P_z|_{T_{\text{step}}/2}^{T_{\text{step}}} = +2.3\,\Delta z_a \text{ with } R^2 = 0.30. \tag{6}$$

We see that a positive $\Delta z_a$ corresponds to a decrease in the first-half vertical impulse and an increase in the second half vertical impulse. In addition to the vertical impulse, the landing leg length is also modulated in response to vertical flight apex deviations. Regressing the leg length $\ell$ at the beginning of stance with the flight apex state, we found that this landing leg length is mostly a function of the vertical position at flight apex:

$$\Delta \ell_{\text{landing}} = 0.3\,\Delta z_a, \text{ with } p < 10^{-4} \text{ and } R^2 = 0.25. \tag{7}$$

Thus, a downward position deviation at flight apex would result in landing with a shorter leg length than nominal (e.g. via flexed knee or ankle). A downward position deviation is analogous to a sudden step-up perturbation, so reducing the landing leg length reduces trip likelihood.

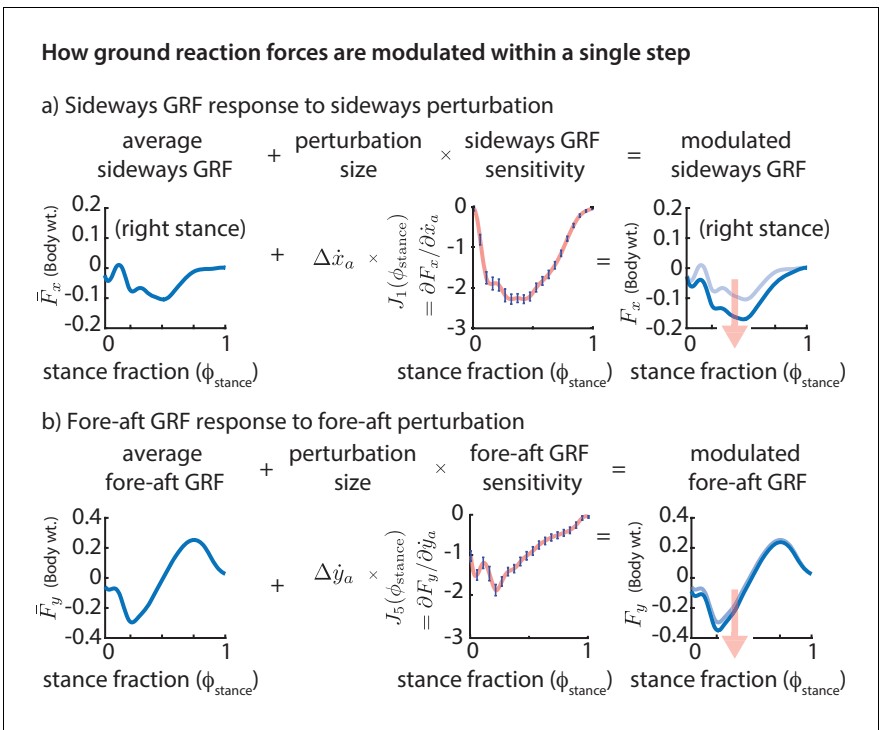

**Figure 5.** Phase-dependent control of GRFs. We show how the GRF components respond to perturbations at the previous flight apex, as estimated by our phase-dependent GRF model. (**a**) Sideways GRF response to a (rightward) sideways velocity perturbation. (**b**) Fore-aft GRF response to a forward velocity perturbation. In both cases, the change in GRF from nominal (shown by the arrow) is obtained as a product of the perturbation size and the sensitivity of the GRF to the perturbation. To produce these plots, we computed a sequence of linear models, predicting the ground reaction forces at a sequence of gait phases through the stance phase, all using the same input variables, namely the flight apex center of mass state. The sensitivities or partial derivatives ($J_1(\phi_{\text{stance}})$ and $J_5(\phi_{\text{stance}})$) shown are the corresponding coefficients in these linear models, plotted as a function of the stance fraction $\phi_{\text{stance}}$ at which the GRF is being predicted by the linear model. In the axis labels, the nominal or average ground reaction forces are denoted with an overbar ($\bar{F}_x$ and $\bar{F}_y$) and the modulated ground reaction forces are denoted without the overbar ($F_x$ and $F_y$).

DOI: https://doi.org/10.7554/eLife.38371.007

## Impulse control is achieved by phase-dependent force modulations

The linear models above tell us how deviations from nominal motion at flight apex are corrected grossly over the next stance. But they do not tell us how the forces are modified continuously throughout a stance phase. The variability of the GRF components $(F_x, F_y, F_z)$ depend on the 'phase' of the stride cycle, specifically, the time fraction $\phi_{\text{stance}}$ of stance (*Figure 2b*). To explain this phase-dependent force variability within a single step, we compute the phase dependent sensitivity of $(F_x, F_y, F_z)$ to the center of mass state as follows. For each output, say $F_x$, we divide the stance duration into 20 phases and compute a linear model for $F_x$ at each of those phases, all with $(\dot{x}_a, \dot{y}_a, z_a)$ as inputs. We refer to the coefficients in these linear models as a function of the phase $\phi_{\text{stance}}$ as the phase-dependent sensitivities of the GRFs (*Figure 5*) to the corresponding predictor variable in $(\dot{x}_a, \dot{y}_a, z_a)$.

The phase-dependent sensitivity of sideways GRF to $\dot{x}_a$ shows that $F_x$ is decreased over the whole step to correct a positive sideways velocity deviation at flight and that a majority of this correction occurs during the middle of stance (*Figure 5a*). Similarly, in response to a positive fore-aft velocity perturbation, the fore-aft GRF is modulated so that there is a net negative force on the body over the next step (*Figure 5b*). The sensitivity of the fore-aft force $F_y$ is more in the first half of stance than during the second half of stance, being modulated more during the deceleration phase (roughly $\phi_{\text{stance}} < 0.5$) than during the acceleration phase (roughly $\phi_{\text{stance}} > 0.5$).

## Foot placement control: step in the direction of the fall

Placing the foot relative to the body allows a runner to modulate the leg force direction and thus the GRF components. The foot position $(x_f, y_f)$ relative to center of mass position at the beginning of stance phase $(x_s, y_s)$ is predicted by the previous flight apex state as described by the following equations. Specifically, sideways foot placement is described by the following equations:

$$\text{Left stance:} \Delta(x_f - x_s) = 0.95 \Delta \dot{x}_a \text{ with } R^2 = 0.64 \text{ and}$$
$$\text{Right stance:} \Delta(x_f - x_s) = 1.00 \Delta \dot{x}_a \text{ with } R^2 = 0.62. \tag{8}$$

The fore-aft foot placement is described by the following equations:

$$\text{Left stance:} \Delta(y_f - y_s) = 0.42 \Delta \dot{y}_a - 0.76 \Delta z_\alpha \text{ with } R^2 = 0.45 \text{ and}$$
$$\text{Right stance:} \Delta(y_f - y_s) = 0.39 \Delta \dot{y}_a - 0.83 \Delta z_a \text{ with } R^2 = 0.46. \tag{9}$$

That is, a sideways velocity perturbation to the body results in the foot being placed further along the direction of the perturbation. So, a rightward perturbation results in a more rightward step. Analogously, a forward velocity perturbation results in the foot being placed further forward relative to the body. As with the impulses, again, there is no significant coupling between sideways and fore-aft directions. Fore-aft foot placement modulation also depends on vertical position deviations, in a manner that the runner lands with a steeper leg when landing from a higher flight apex $z_a$. Such dependence of landing leg angle on vertical position is analogous to behavior in terrain-change experiments (*Daley and Biewener, 2006*; *Müller et al., 2012*; *Qiao and Jindrich, 2012*; *Birn-Jeffery and Daley, 2012*), as discussed in detail later. We speculate that using foot placement based on center of mass state may be an efficient way to affect the center of mass motion, compared to, say, changing the leg force magnitudes and leg joint torques after the foot touches down (*Clark, 2018*).

## Swing foot re-positioning happens during flight, just before foot touchdown

One possibility is that the foot placement deviations are achieved early on during the swing phase and this deviation is preserved during swing until the foot touchdown. However, this does not appear to be the case. *Figure 6* shows the fraction of foot placement variance predicted by the swing foot state over the previous step. Less than 10% of the eventual foot placement is predicted by the swing foot at the beginning of flight phase (*Figure 6*). The explanatory power of the swing foot rises rapidly during the flight phase from less than 10% to a 100% when it becomes the next stance foot, suggesting that most swing foot re-positioning may happen during this flight phase.

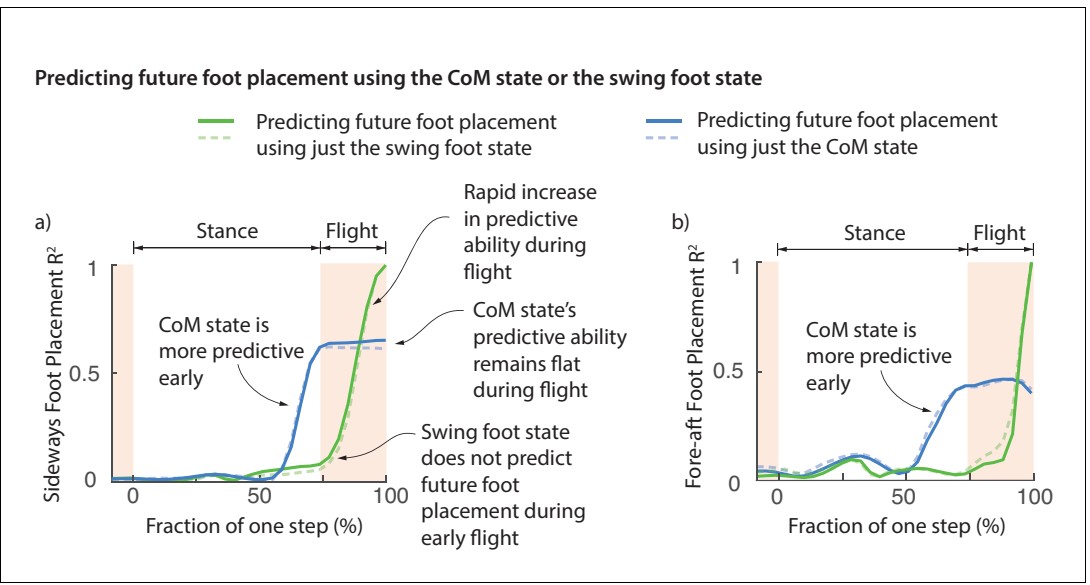

**Figure 6.** Swing foot control before foot placement. The fraction of sideways foot placement (panel a) and fore-aft foot placement (panel b) variance at beginning of stance predicted by the center of mass (CoM) state or swing foot state during the previous one step (flight and stance). To produce this figure, a sequence of linear models were built for predicting the foot placement based on the center of mass state or swing foot state during different phases through the previous step. We plot the $R^2$ value corresponding to these linear models (that is, fraction of variance explained) as a function of the gait phase used for the prediction; the gait phase is represented as the fraction of a step starting from beginning of previous stance phase. The solid and dashed lines represent right and left foot placements, respectively.

DOI: https://doi.org/10.7554/eLife.38371.008

## Center of mass state predicts future foot placement before the foot state does

At the beginning of flight phase (and earlier), the center of mass state is a vastly better predictor of the next foot placement than the swing foot itself (*Figure 6*). We can predict the foot placement using the center of mass state better than just the relative swing foot state until about 100 ms before foot touchdown. The explanatory power of the center of mass remains flat during flight. This flatness is likely because center of mass state follows a parabolic path during flight and thus accumulates no new variation. This lag between the explanatory power of the center of mass and the foot suggests that the error information in the center of mass state is yet to be fully incorporated into the swing foot re-positioning until the flight phase. During the brief flight phase, when the swing foot's explanatory power increases, information from center of mass state is transferred to the foot, presumably via some mixture of feedback control and feedforward dynamics.

## Continuous stance state feedback, station-keeping, and running speed do not significantly affect stance control

As an alternative to control based on discrete monitoring of deviations at the previous flight apex state, we considered a 'continuous control' model. Specifically, we obtained linear models for the GRFs based on the current center of mass state during stance $(\dot{x}, \dot{y}, z)$. These linear models did not differ significantly in the fraction of GRF variance explained, compared to the apex-based control model ($p = 0.94$). In the linear models above, adding the sideways and fore-aft apex body position $(x_a, y_a)$ to the explanatory variables improves the $R^2$ values by less than $1.5\%$. Thus, the runners did not prioritize controlling their position relative to the treadmill (station-keeping). Further, the regression coefficients for $(\dot{x}_a, \dot{y}_a, z_a)$ did not vary significantly across the three running speeds ($p > 0.05$, paired t-test).

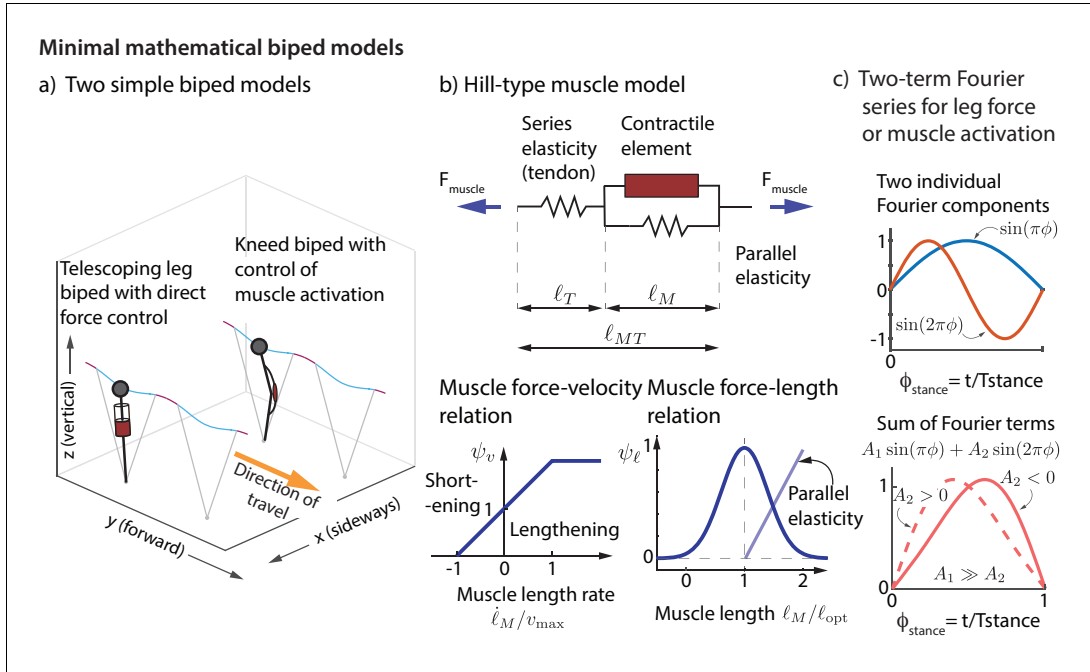

**Figure 7.** Minimal mathematical biped models. (**a**) Two simple biped models were simulated: a telescoping leg model with direct force control and a kneed biped with activation control of the muscle at the knee. (**b**) The muscle in the second model is a classic Hill-type muscle (*Zajac, 1989*), composed of an active contractile element, a series elastic element (tendon), and a parallel elastic element. The force $F_{CE}$ in the active contractile element for the muscle model depends on the muscle length $\ell_m$ through a force-length relationship $\psi_\ell$, on the muscle length rate $\dot{\ell}_m$ through a force-velocity relation $\psi_v$, and the activation $a$, so that $F_{CE} = aF_{iso}\psi_v\psi_\ell$, where $F_{iso}$ is the maximum isometric force in the muscle (*Zajac, 1989*). (**c**) The control input for both models is represented as a Fourier sum of two sine waves of frequencies $T_{stance}^{-1}$ and $2T_{stance}^{-1}$. For the first model, the force is represented by this Fourier sum and for the second model, the muscle activation. This two-term function is able to allow leg force profiles without a time-reversal symmetry as shown (force peak not occurring at mid-stance).
DOI: https://doi.org/10.7554/eLife.38371.009

## Approximate left-right symmetry in the control

The running control gains have approximate bilateral (left-right) symmetry. The gains that couple sideways direction variables and either fore-aft or vertical direction variables have mirror-symmetry (see *Equations 1,2,8,9*). That is, these gains for the left leg's stance are the negatives of corresponding gains for the right leg's stance. On the other hand, gains that couple one sideways variable with another sideways variable, or one fore-aft variable with another fore-aft variable, are the same for the left and right legs without any such sign changes. This mirror symmetry in running control likely follows from the approximate mirror symmetry in body physiology about the sagittal plane and was also found in walking (*Wang and Srinivasan, 2014*; *Ankaralı et al., 2015*). This symmetry suggests the lack of a substantially dominant limb for running control, in contrast to the asymmetry and limb role differentiation that occurs in some other tasks (*Peters, 1988*).

## Simple models of running capture GRFs and phase-dependent GRF modulations

We now show that the experimentally derived control strategies described above are sufficient to control the running dynamics of a simple mathematical model of a biped. We consider a biped with point-mass upper body and massless telescoping legs capable of generating arbitrary force profiles (unlike a spring). We considered two versions of this biped model (*Figure 7a*), one with direct control of the leg force and another that produces leg forces via Hill-type muscles (*Figure 7b*). See Materials and methods for how the nominal running motion and the feedback controllers are specified for the models.

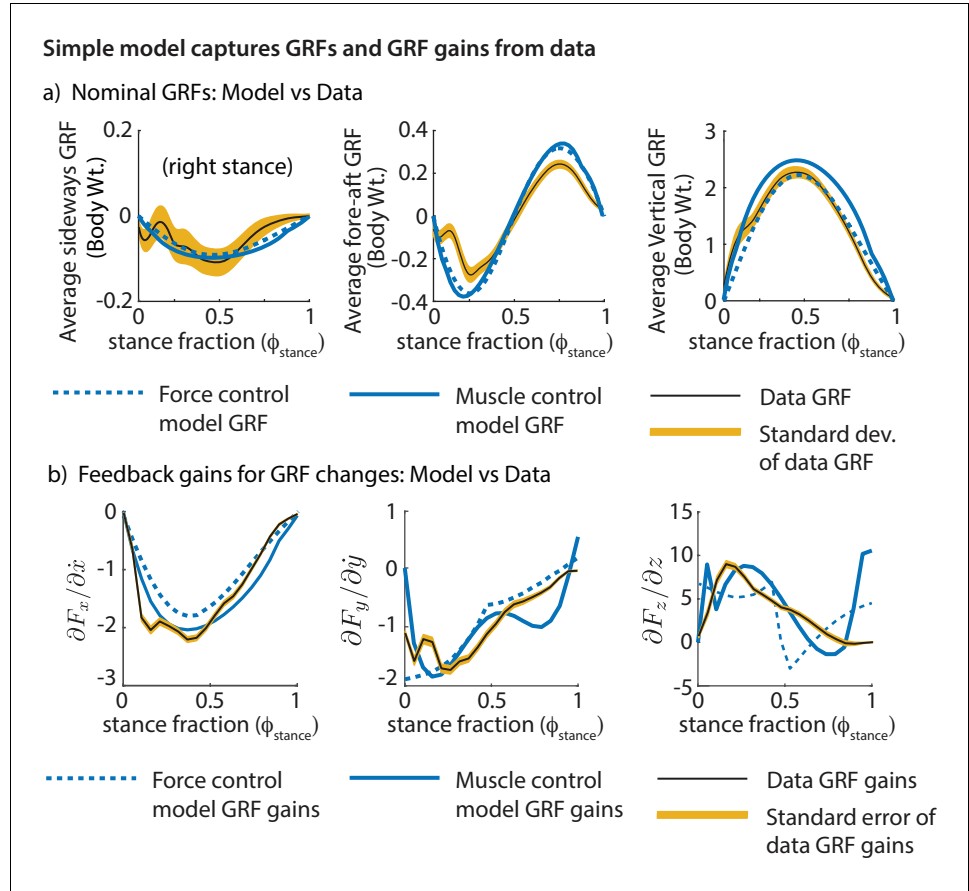

**Figure 8.** Comparing the simple biped models and human running data. Both models fit the experimental GRFs and feedback gains reasonably well, despite not having been made to match them explicitly: (**a**) Mean GRFs over stance in three directions. GRFs are reported as a fraction of body weight. (**b**) Phase-dependent feedback gains describing the sensitivity of the sideways GRF to sideways velocity perturbation, fore-aft GRF to a fore-aft velocity perturbation, and the vertical GRF to vertical position perturbation at the previous flight apex. The standard deviations of the experimentally derived curves are shown as yellow bands.
DOI: https://doi.org/10.7554/eLife.38371.010

We find that the models' ground reaction forces are similar to experimental data despite not explicitly matching the curves (*Figure 8a*). Further, we find that the phase-dependent ground reaction force feedback gains for the models are qualitatively similar to the phase-dependent gains inferred from experiment (*Figure 8b*), again, despite not explicitly fitting the shape of these phase-dependent gains. This shows that these simple models can not only capture the average motion during running, but also how the runner responds to deviations from the average motion.

## Human-derived controller stabilizes a minimal model of bipedal running

The simple models' running motions are not stable without the controller: an arbitrarily small perturbation makes it diverge from the original running motion. With the foot placement and leg force controller turned on, the running motion is asymptotically stable. *Figure 9a–d* shows the model recovering from fore-aft, sideways, and vertical perturbations at flight apex. It is a mathematical theorem that a stable periodic motion that can reject perturbations at one phase (say, flight apex) can reject perturbations at any phase (*Guckenheimer and Holmes, 2013*). So, it follows that our model rejects perturbations at any phase.

The inputs to the feedback controller $(\dot{x}_a, \dot{y}_a, z_a)$ do not include the absolute sideways and fore-aft position $(x_a, y_a)$ of the runner. Therefore, the controller does not correct position perturbations (station-keeping). A sideways or fore-aft push to the model results in convergence to the nominal running motion, except for a sideways or fore-aft position offset (*Figure 9c*).

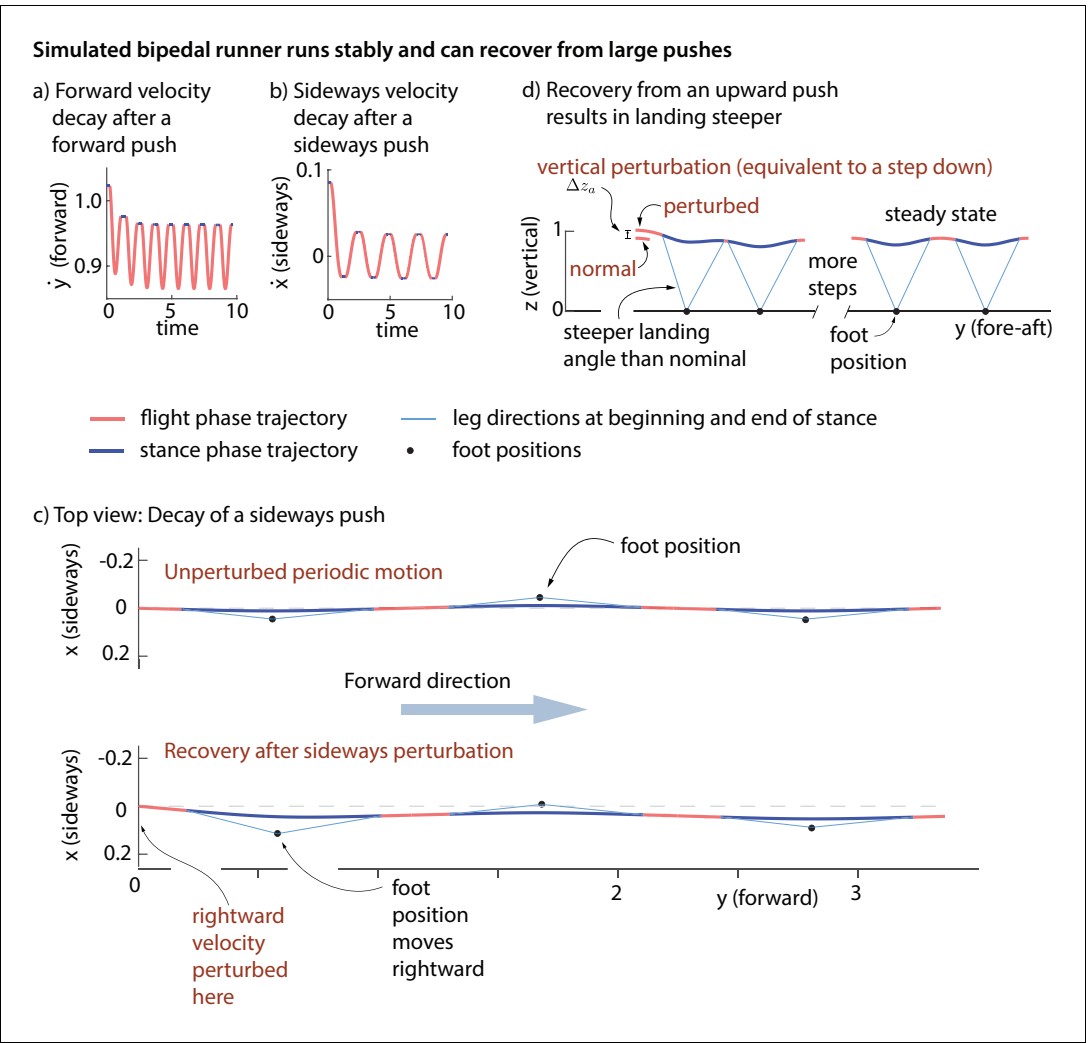

**Figure 9.** Stability of running in the simple biped model using the human-derived controller. We illustrate the stability of the running model by showing how large perturbations at flight apex decay. (a) Decay of a forward velocity perturbation. (b) Decay of a sideways velocity perturbation. (c) Top view center of mass trajectory, showing the unperturbed running motion and a running motion that recovers from a rightward velocity perturbation. The rightward perturbation elicits a rightward foot placement, compared to the nominal foot placement during unperturbed running. (d) Sagittal view of running, recovering from an upward position perturbation. On the first step, the leg touch-down angle is steeper than the touch-down angle during unperturbed running and the contact time is shorter. All quantities are non-dimensional. See *Videos 1,2* and *3* for illustrative animations of the biped model recovering from forward, sideways, and vertical perturbations, respectively.
DOI: https://doi.org/10.7554/eLife.38371.011

## Non-zero leg work for energy-changing perturbations

*Figure 10* illustrates the leg work-loop for the unperturbed run (net zero work) and when positive perturbations are applied to sideways and fore-aft velocities, and vertical positions. All such positive perturbations result in net negative work on the first step after the perturbation, reflected in the work-loops with net negative area within them. Such net positive or negative leg work is clearly necessary to recover from perturbations that change the total mechanical energy of the runner, as was recognized in prior discussions of the energy-neutral spring-mass model of running (*Ghigliazza et al., 2005*; *Biewener and Daley, 2007*; *Srinivasan and Holmes, 2008*).

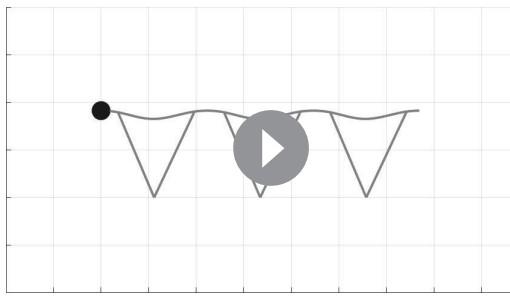

**Video 1.** Animation of point-mass biped model with the human-derived controller recovering from a forward velocity perturbation.
DOI: https://doi.org/10.7554/eLife.38371.012

## Explaining variability: muscle-driven running model does not fall despite noise

To simulate the step-to-step variability in real human running, we added 'noise' to our foot placement and leg forces (for the direct force control model) or muscle activations (for the muscle-driven model) and simulated the biped models for a few hundred steps. This noise is meant to model the phenomenon that intended muscle forces tend to deviate from actual muscle forces due to motor noise (*Harris and Wolpert, 1998*). We find that while the direct leg force control model falls down, the runner with muscles does not fall down for hundreds of steps despite the noise. The stable motion of the center of mass in the presence of noise-like perturbations is shown in *Figure 11a*. The variability in the center of mass state at flight apex for the model (*Figure 11b*) as a result of the simulated noisy control is qualitatively similar to the variability found in experiment (*Figure 2a*). The model is also able to run without falling despite vertical position perturbations at flight apex, which are equivalent to uneven terrain. Thus, even though the model was derived using data on horizontal ground, it is capable of running robustly on uneven terrain. The muscle-driven model is robust to motor noise presumably because of the intrinsic stabilizing properties of force-length and force-velocity relations (*Hogan, 1984*; *Jindrich and Full, 2002*).

## Discussion

We have mined the step-to-step variability in human running to show how humans modulate leg forces and foot placement to run stably. We then used these data-derived control strategies on a biped model, demonstrating robustness to discrete perturbations and persistent motor noise.

We have shown that humans use foot placement or leg angle control in a manner that they step in the direction of the perturbation, thereby directing the leg force so as to oppose the perturbation. This result provides an empirical basis for ad hoc assumptions about leg angle control made in previous running models (*Seyfarth, 2003*; *Ghigliazza et al., 2005*; *Peuker et al., 2012*). The foot placement controller derived from running data is qualitatively similar to the classic Raibert-like controller used in early running robots (*Raibert, 1986*) in that the foot placement opposes velocity deviations with no sideways-fore-aft

**Video 2.** Animation of point-mass biped model with the human-derived controller recovering from a sideways velocity perturbation.
DOI: https://doi.org/10.7554/eLife.38371.013

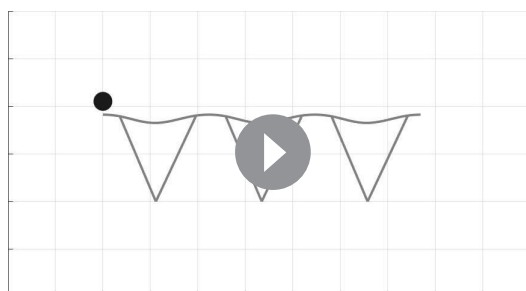

**Video 3.** Animation of point-mass biped model with the human-derived controller recovering from a vertical position perturbation.
DOI: https://doi.org/10.7554/eLife.38371.014

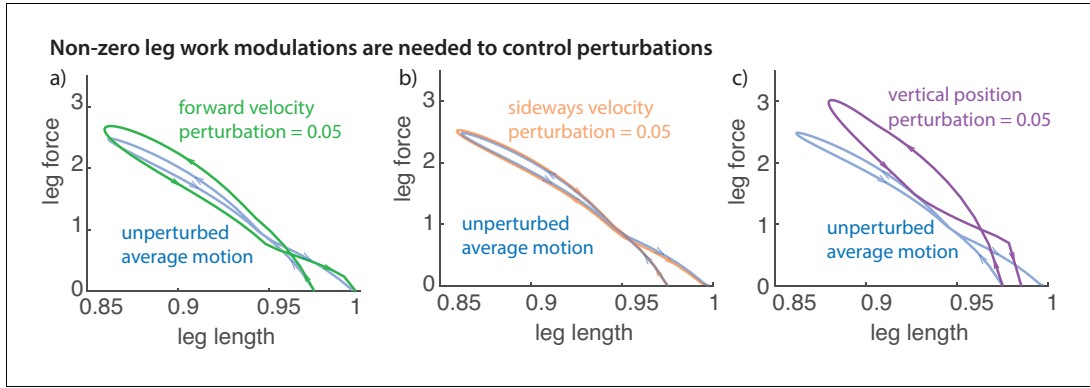

**Figure 10.** Work loop and net leg work. Work loops for the muscle-driven biped model without perturbations and with perturbations in the (**a**) fore-aft velocity, (**b**) sideways velocity, and (**c**) vertical position. The work loop plots force versus leg length and the signed area included in it is the net work performed by the leg.
DOI: https://doi.org/10.7554/eLife.38371.017

coupling, but differs in that it has a dependence on vertical position perturbations. This makes such robotic controllers inadvertently biomimetic. Humans use similar foot placement control in walking, stepping in the direction of the perturbation (*Hof et al., 2010*; *Wang and Srinivasan, 2014*). Previous work had shown that appropriate foot placement is used in running ostriches while turning (*Jindrich et al., 2007*), running humans in cutting maneuvers (*Besier et al., 2003*), and turning while walking (*Patla et al., 1999*).

Some past work on inferring stability from variability focused on kinematic measures of variability such as Floquet multipliers, finite-time Lyapunov exponents (*Dingwell et al., 2001*) and long term correlations in walking and running variability (*Hausdorff et al., 1995*; *Jordan et al., 2006*; *Kaipust et al., 2012*). Such measures can provide discriminative diagnostic measures (*Kaipust et al., 2012*), but do not attempt to provide a causal narrative about how locomotion is controlled. Our approach here, rooted in Newton-Euler mechanics, is able to discover potential causal strategies underlying locomotion stability, and by extension, could inform treatment of pathological unstable movements in addition to diagnosis. Other past studies have used variants of the principal component analysis (*Cappellini et al., 2006*; *Maus et al., 2015*) to demonstrate that the intrinsic variability in human locomotion may reside in a lower dimensional manifold (*Cappellini et al., 2006*; *Chang et al., 2009*; *Yen et al., 2009*; *Dingwell et al., 2010*; *Maus et al., 2015*). Here, by focusing on how the center of mass is controlled through forces, we have implicitly used a physics-based dimensionality reduction to examine the dominant control strategies.

While our work relies on linear regressions from data, the basic physics relating the inputs and outputs in these models suggest a natural causal account. This causal account, based on simplifying modeling assumptions, ignores the effect of variables not considered here. Our goal here was to delineate the explanatory power of controller descriptions based on center of mass state. To identify the effect of perturbations of other possibly relevant state variables (such as trunk attitude and angular velocity), we may need to either independently perturb these state variables or show that the natural variability in such variables is not significantly correlated with the center of mass state.

The gain relating sideways foot placement and sideways velocity deviation was about 2.5 times greater than the gain relating fore-aft foot placement and fore-aft velocity deviation; a similar factor of 3 was found in walking (*Wang and Srinivasan, 2014*), perhaps reflecting the greater sideways instability of a biped without foot placement control (*Ghigliazza et al., 2005*). Also consistent with lower control authority and a greater fall propensity in the sideways direction, we find that the recovery from a sideways perturbation is faster than from a fore-aft perturbation. While station-keeping was not prioritized over a single step, it may occur on a slower time-scale with a multi-step controller, not considered here.

The results we have presented have been for data pooled over all subjects. Performing the regressions for data from individual subjects indicates that the dominant terms in the inferred controllers are similar for all subjects; the subject-to-subject variability in the estimated control gains are shown in *Figure 12*. *Figure 12* illustrates how the accuracy of an estimated control gain depends on

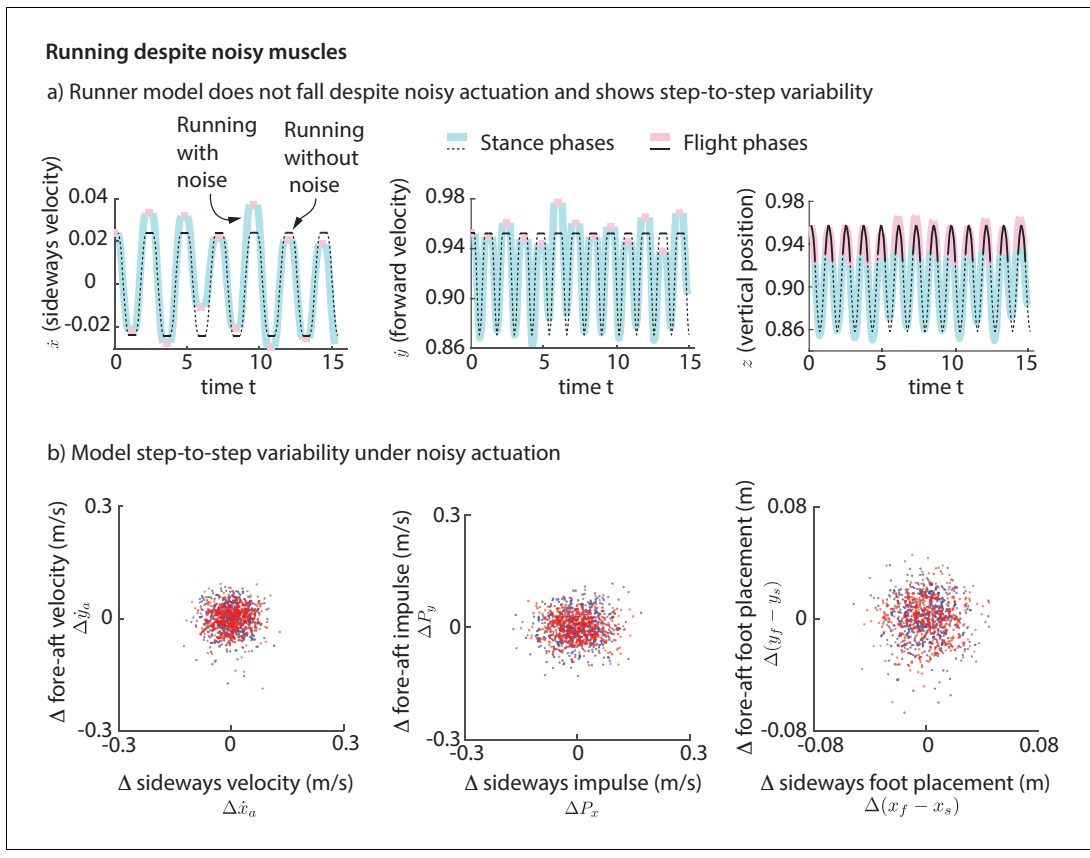

**Figure 11.** Running with noise. (a) Multiple steps of the biped model running in the presence of noisy foot placement and muscle activations (blue for stance phases and pink for flight phases). Periodic nominal motion in the absence of noise (black, solid lines for stance, dashed for flight). (b) Deviations in flight apex state, GRF impulse, and foot placement from nominal for a 1000 steps (500 left steps as blue dots, 500 right steps as red dots), showing behavior analogous to *Figure 2*. This variability is not explicitly specified, but instead emerges from the interaction between the motor noise and the controlled dynamics.

DOI: https://doi.org/10.7554/eLife.38371.015

the number of strides used for regression. For such linear regressions, the error estimate (standard deviation) is expected to decrease with $N_{\text{stride}}$ like $1/N_{\text{stride}}^2$, so that a factor of 10 decrease in error requires a 100-fold increase in sample size (*Wang and Srinivasan, 2012*; *Hamilton, 1994*). This dependence on $N_{\text{stride}}$ may guide selection of sample sizes for future experimental designs.

Our model predicts that when a runner starts at a higher-than-normal height at flight apex, or equivalently, encounters a step-down, the runner lands with a steeper leg angle (*Figure 9d*). Such behavior has been observed in humans and bipedal running birds running with large unforeseen or visible step-downs (*Daley and Biewener, 2006*; *Grimmer et al., 2008*; *Müller et al., 2012*; *Qiao and Jindrich, 2012*). Conversely, step-ups decrease touch-down angle, as predicted (*Birn-Jeffery and Daley, 2012*). This behavior has been attributed to swing leg retraction just before foot contact (*Seyfarth, 2003*), but our foot placement controller captures this phenomenon despite not having explicit leg swing dynamics. While the terrain perturbations in the aforementioned experiments were large (5–20 cm), our model is based on data with tiny step-to-step deviations (vertical position $z_a$ s.d. 5 mm). This agreement indicates that humans may use qualitatively similar control strategies for large external perturbations and small intrinsic perturbations. Such foot placement control has also been used to control robots running on uneven terrain (*Hodgins and Raibert, 1991*).

It is expected that any running controller that achieves asymptotic stability will need to perform net mechanical work in response to perturbations that decrease or increase the body's mechanical energy (*Ghigliazza et al., 2005*; *Srinivasan and Holmes, 2008*; *Maus et al., 2015*). Our results are consistent with such expectation, as illustrated by the work-loops with net mechanical work in

*Figure 10.* Energy-conservative spring-like leg behavior does not allow such net mechanical work and can achieve only partial asymptotic stability, not being able to handle energy-changing perturbations (as noted by (*Ghigliazza et al., 2005*)). Indeed, it is generally thought that even the spring-mass-like steady state center of mass motion in running is due to considerable muscle action and has been termed pseudo-elastic (*Ruina et al., 2005*) or pseudo-compliant (*McN. Alexander, 1997*). Remarkably, energy-optimal running movements in models with no leg springs produce similar spring-mass-like center of mass trajectories (*Srinivasan, 2011*), with leg muscles performing equal amounts of positive and negative work.

A previous article (*Maus et al., 2015*) fit running data to variants of the spring-mass model, allowing the spring stiffness and spring length to change during stance, and showing that constant values for these parameters cannot fit running data. Here, we have used a simpler model to directly describe the control of stance leg force or activation (*Figure 8*). Such direct control of leg force or activation is perhaps more parsimonious than the simultaneous control of two variables, namely, spring stiffness and length. We have shown that humans modulate GRF continuously over the whole stance phase for control (*Figure 5*); Maus and colleagues (*Maus et al., 2015*) assumed, for simplicity, an instantaneous finite energy input at mid-stance.

The stabilizing responses we have characterized in this study are likely due to a mixture of feed-forward dynamics and active neurally mediated feedback control. When we use the term "control" here, we implicitly refer to this mixture. It is hard to rigorously separate the roles of feedforward and feedback control without recording motor neuronal outputs and how these outputs interact with the properties of muscles. Nevertheless, we can determine the feasibility of feedback control by checking whether there is enough time for feedback control, given typical neuromuscular latencies. Our typical flight phase durations are greater than or about roughly equal to the typical short- or middle-latencies in reflex or feedback loops involving vestibular (*Fitzpatrick et al., 1994*; *Iles et al., 2007*) or proprioceptive mechanisms (*Pearson and Collins, 1993*; *Sinkjær et al., 1999*). This suggests

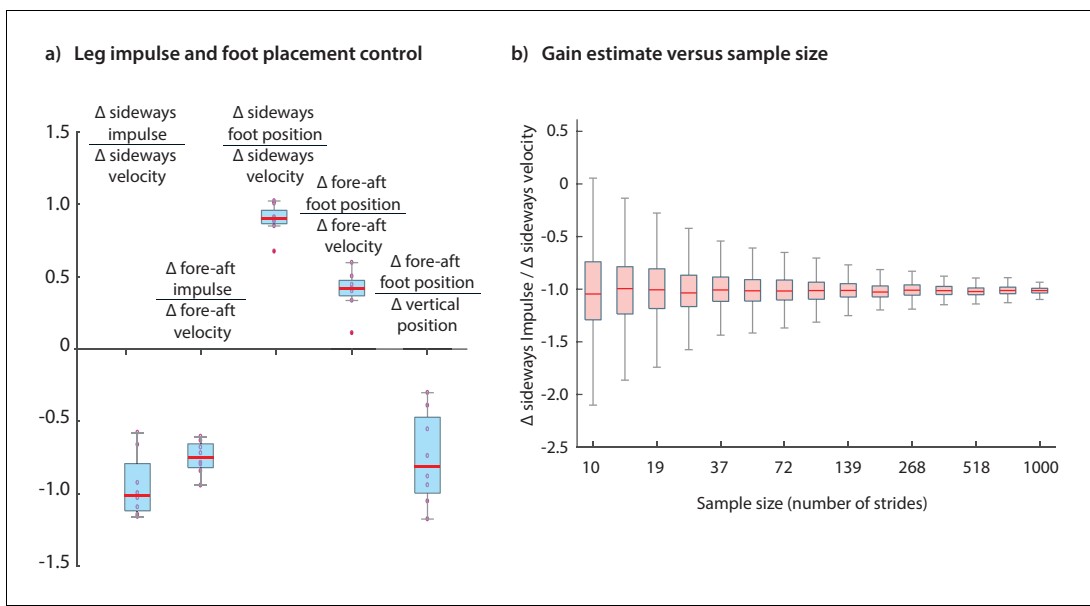

**Figure 12.** Variability in control gains due to subjects and sample sizes. (a) The coefficients in the impulse control equations (*Equations 1-2*) and foot placement equations (*Equations 8-9*). The box plots indicate distributions of coefficients over all subjects; coefficients for each subject are also shown (red circles). (b) The dependence of coefficient estimate on the sample size, namely the number of strides $N_{stride}$ used. The box plot indicates variability in control gain estimates. This plot was generated using bootstrap statistics (*Efron and Tibshirani, 1994*) by resampling from the pooled data from all subjects, and plotting the distribution of control gains obtained for each $N_{stride}$ over multiple samples. This graph corresponds to the left stance impulses, but the corresponding graph for the right stances is nearly identical. Other coefficients exhibit similar trends with sample size. The boxes in the box plots show the median and the 25th to 75th percentile and the whiskers indicate all data within 2.7 standard deviations from the median.
DOI: https://doi.org/10.7554/eLife.38371.016

feasibility of feedback based on flight phase or late stance phase information regarding center of mass state. While we have focused on the control of stance based on the previous flight apex, we have found that equivalent controllers based on the center of mass state at the end of previous stance have similar predictive ability (*Figure 6*), thus allowing more time for neural feedback. Specifically, the lag between the information in the center of mass state and the swing foot state regarding future foot placement is about 0.1 s for sideways placement and about twice that for fore-aft foot placement, suggesting sufficient time for neurally mediated feedback control of foot placement (*Figure 6*).

Center of mass state or other body state information needed for feedback control could be estimated by the nervous system by integrating sensory signals from vision (*Patla, 1997*), proprioceptive sensors (especially when the foot is on the ground (*Sainburg et al., 1995*)), and vestibular sensors (*Angelaki and Cullen, 2008*), potentially in combination with predictive internal models (*Wolpert et al., 1995*; *Cullen, 2004*). In future work, repeating the calculations herein (for instance, *Figure 6*) for experiments that systematically block or degrade (say, by adding sensory noise) one or more of these sensors may tell us the relative contributions of these sensors to running control. We speculate that most available relevant sensory information is used, perhaps analogous to an optimal state estimator (*Kuo, 2005*; *Srinivasan, 2009*), and degrading one sensor may result in sensory re-weighting on a slow time-scale (*Carver et al., 2006*; *Assländer and Peterka, 2014*). Such experiments may also help explicitly distinguish the effects of sensory and motor noise, which we have implicitly combined here into a single residual term in the linear regressions.

In this work, we have obtained a running controller with simplifying assumptions. Because humans have extended feet, non-point-mass upper bodies and legs with masses, the simple point-mass model may not capture all aspects of the running data (*Bullimore and Burn, 2006*; *Srinivasan and Holmes, 2008*). Further, we have made simplifying assumptions about muscle architecture, muscle properties (linear force-velocity relations), and muscle activation, which are meant to capture the main qualitative dynamical features of muscles, rather than model them quantitatively precisely. For instance, we used a linear force-velocity relation, which may be sufficient to produce damping-like and stabilizing muscle behavior when activated, but this damping behavior may be accentuated by a more realistic nonlinear force-velocity relation.

Future work will also involve obtaining controllers for more complex biped models and muscle models with different control architectures, which, for instance, might include feedback control based on not just the center of mass state, but the states of individual body segments. We have focused on linear relations between state deviations and control, as this is naturally suited for small deviations and perturbations that our data contains. In future work, we hope to examine the range of perturbation sizes for which this linear description is accurate by comparing this linear control to responses in experiments with larger perturbations, also inferring nonlinear descriptions should they improve predictive capability. We will also examine other control architectures, for instance, more explicitly incorporating state estimation and considering continuous control of motor outputs based on an estimated state, partly correcting for neural latencies using internal models.

The methods used here are simple and non-invasive: they can be replicated to study running stability and control in other animals, or indeed, other approximately periodic tasks such as flapping flight and swimming. These methods are suitable for analyzing differences in different populations like athletes and non-athletes, the young and the elderly, and adults with and without movement disorders. Once such differences are well-characterized, this information could be used, say, in a rehabilitation setting to track progress from a controller in the presence of a movement disorder to a more healthy controller, and to design rehabilitation robots that assist in this progress.

## Materials and methods

We collected running data by conducting human subject experiments, obtained linear models to characterize the control strategies hidden in the step-to-step variability, and performed dynamic simulations using the inferred controller on a mathematical model of the runner.

### Experimental methods

The protocols were approved by the Ohio State University Institution Review Board and subjects participated with informed consent. Eight subjects, three female and five male (age 25.0 ±5 years,

weight 66.8 ±7 kg, height 1.8 ±0.14 m, leg length 1.05 ±0.08 m, mean ± s.d.) ran on a split-belt treadmill at three constant speeds: 2.5, 2.7 and 2.9 m/s, presented in random order. Each speed had 2075 ± 67 strides across all subjects (one stride = two steps) with subjects running about 3.5 min on average. Subjects wore a loose safety harness that did not constrain their motion. Three-dimensional ground reaction forces and moments on each belt of the treadmill were recorded by separate six-axis load cells (Bertec Inc, 1000 Hz). Body segment motion was measured using marker-based motion capture (Vicon T20, 100 Hz) with four reflective markers on each foot and on the torso.

## Calculating input state variables during flight apex

We define flight apex as when the center of mass velocity reaches its peak height ($\dot{z} = 0$). The input to the running controller is drawn from the center of mass state at flight apex, namely position $(x_a, y_a, z_a)$ and velocity $(\dot{x}_a, \dot{y}_a, \dot{z}_a)$. Unless otherwise specified, we use the flight apex state $(\dot{x}_a, \dot{y}_a, z_a)$ as inputs in our linear models. The vertical velocity $\dot{z}_a$ at flight apex is zero by definition and hence not included as an input. The center of mass velocities are obtained by integrating the center of mass accelerations, that is, the mass-normalized net force on the body: $\ddot{x} = F_x/m$, $\ddot{y} = F_y/m$, and $\ddot{z} = F_z/m - g$, where $F_x, F_y$ and $F_z$ are the measured ground reaction forces on the body. To obtain the integration constants, we assume that the mean velocity and acceleration over the whole trial are zero, because the person does not translate appreciably in the lab frame over a trial. To remove the slow integration drift in the center of mass velocity, we used a high-pass filter with a frequency cut-off equal to an eighth of mean step frequency (*Luinge and Veltink, 2005*; *Schepers et al., 2009*). Changing this high-pass filter cut-off to a twentieth of the step frequency instead, or using a piecewise-linear de-trending over 20 steps, do not affect any of this article's con- clusions. This is because the stability-critical time-scales are much shorter. We ignored air drag here, because including it changed the velocities by less than $10^{-5}$ ms$^{-1}$, which is much smaller than the step-to-step variability. We use a weighted mean of four markers, roughly at the sacral level, as an approximation of the center of mass position (*Gard et al., 2004*; *Wang and Srinivasan, 2014*; *Perry and Srinivasan, 2017*).

## Calculating the output control variables during stance

We assume that the following variables are used to control the runner: GRFs, foot placement, and the landing leg length. Stance phases are identified as when the vertical GRF exceeds a threshold value to account for measurement noise ($F_z > 30$ N). The corresponding stance duration is $T_{\text{stance}}$. The GRF impulses $(P_x, P_y, P_z)$ for each step are obtained by integrating the GRF components over the stance phase ($P_x = \int_0^{T_{\text{stance}}} F_x \, dt$, etc). In addition to considering how GRF control occurs grossly over one step, we also consider GRF control throughout stance as a function of stance phase fraction $\phi_{\text{stance}}$. Each stance phase is divided into $n$ bins of duration $T_{\text{stance}}/n$. To approximate how the GRFs changes with the stance phase fraction $\phi_{\text{stance}}$, we used the binned averages of the GRF in each of $n = 20$ bins.

## Linear regressions between the outputs and the inputs

We compute the mean values of the inputs over all steps in each trial and obtain deviations from these means ($\Delta\dot{x}_a, \Delta\dot{y}_a, \Delta z_a$). Similarly, we compute the deviations from the means of the output varia- bles $\Delta F(\phi_{\text{stance}})$, $\Delta P$, and $\Delta(x_f - x_s, y_f - y_s)$. We use ordinary least squares regression to obtain linear models between the inputs and the outputs and report significant coefficients. Specifically, we have $\Delta\text{Output} = J \cdot \Delta\text{Input}$, where the Jacobian matrix $J$ represents the matrix of coefficients in the linear model. Each element of the matrix $J$ quantifies the sensitivity of an output variable to small changes in a corresponding input variable, as inferred from the data and subject to the simplifying model assumptions. These sensitivity coefficients could be interpreted as partial derivatives, such as: $\partial T_{\text{stance}}/\partial\dot{x}_a$, $\partial F_y(\phi_{\text{stance}})/\partial\dot{y}_a$, and so on. Unless otherwise specified, the results presented are based on deviations of all subjects pooled together as one dataset, but we find that the models of individ- ual subjects' data are qualitatively similar (as indicated in *Figure 12*). The coefficients for the right leg and left leg are computed separately, to accommodate sign changes due to symmetry about the sagittal plane.

## Regressions with phase-dependent inputs

In addition to the regressions described above using the flight apex state as the predictor, we used the center of mass state $(\Delta\dot{x}_a, \Delta\dot{y}_a, \Delta z_a)$ at different phases over the previous step to predict each of the stance phase outputs. Specifically, for each stance phase output, we performed $n = 20$ regressions, each using the center of mass state at one of the $n = 20$ equally spaced gait phases over the previous step, where one full step is defined as starting and ending at a touchdown. This analysis allows us to investigate the predictive ability of the center of mass state at different phases. For these phase-dependent regressions, in addition to using the center of mass state as the predictor, we repeated the calculations using the swing foot state (position and velocity relative to the center of mass), so as to compare the different predictive abilities as in *Figure 6*.

## Implementing the data-derived control on a minimal mathematical biped

We consider two simple models of running, similar in spirit to previous models in terms of simplicity (*Blickhan and Full, 1993*; *Geyer et al., 2006*; *Srinivasan and Holmes, 2008*), but generalized such that the leg forces are not constrained by ad hoc spring-like-leg assumptions (*Srinivasan, 2011*). Instead, the biped controller details are inferred from our experimentally obtained linear models. Both biped models have a point-mass upper body and massless legs (*Srinivasan and Ruina, 2006*; *Srinivasan, 2011*), that can change effective leg length during stance by modulating the leg force (*Figure 7a*). During flight phase, the point-mass body undergoes parabolic free flight. The legs can apply forces on the upper body during stance phase. The two models, dubbed 'direct force control model' and 'muscle control model' differ in how the leg force is produced and controlled. For the muscle control model, we use a Hill muscle model with force-length and force-velocity relations (*Figure 7b, c and d*). The 3D equations of motion of the point-mass biped are: $m\ddot{x} = F_{\text{leg}} \cdot (x - x_{\text{foot}})/\ell$, $m\ddot{y} = F_{\text{leg}} \cdot (y - y_{\text{foot}})/\ell$, and $m\ddot{z} = -mg + F_{\text{leg}} \cdot (z - z_{\text{foot}})/\ell$, where $F_{\text{leg}}$ is the scalar leg force, $(x_{\text{foot}}, y_{\text{foot}}, z_{\text{foot}})$ is the foot position with $z_{\text{foot}} = 0$ on flat terrain and $\ell = \sqrt{(x - x_{\text{foot}})^2 + (y - y_{\text{foot}})^2 + (z - z_{\text{foot}})^2}$ is the leg length from body to foot. In the 'direct force control model', the object of control is the leg force $F_{\text{leg}}$ during stance phase. In the 'muscle control model', the object of control is the muscle activation $a_{\text{muscle}}$, which is converted to muscle force via the force-length and force-velocity equations of Hill-type muscles (*Figure 7a–b*). See (*Zajac, 1989*; *Srinivasan and Ruina, 2006*; *Srinivasan, 2011*) for more detailed equations of motion and muscle model equations.

Both models have two terms in their control: (1) a feedforward or 'nominal' term, that depends only on the average or desired periodic motion and (2) feedback modification of the control in response to state deviations at flight phase. The model's leg force or muscle activation is modeled as a two-term sine series of the form $A_1 \sin(2\pi t/2T_{\text{stance}}) + A_2 \sin(2\pi t/T_{\text{stance}})$, as shown in *Figure 7e*. By changing the relative weights of $A_1$ and $A_2$, the shape of the leg force profile can be changed from being symmetric about the peak force to being asymmetric, with the peak force preceding or following mid-stance. We parameterize the running motion using stance duration $T_{\text{stance}}$, flight duration $T_{\text{flight}}$, 2D foot placement $(x_{\text{foot}}, y_{\text{foot}})$, 3D initial conditions for stance $(x(0), y(0), z(0))$, and the coefficients of the two-term sine series ($A_1$ and $A_2$). We solve for these variables to obtain a periodic running motion that accurately match the forward speed, step period, step width, and peak leg force from experimental data (*Figure 8a*) by using an optimization procedure (*Srinivasan and Holmes, 2008*; *Srinivasan, 2011*) that enforces a constraint satisfaction tolerance of less than $10^{-6}$. The runner leaves the ground when it reaches the maximum leg length, but the nominal leg length at landing is assumed to be shorter (95%) than the maximum leg length, as seen in running data (*Voloshina and Ferris, 2015*). We enforce that left and right stances are mirror symmetric. Unlike previous simple running models, our model's nominal periodic motion has non-zero step width and a stance phase that is asymmetric about mid-stance. This asymmetric stance is due to unequal landing and take-off leg lengths, and the asymmetry of the leg force or muscle activation about mid-stance.

The foot placement control for the models are based on the experimentally derived control and given by the linear model in *Equations 8 and 9*. The leg force feedback control based on apex body state, for the direct force control model, has gains as shown in *Figure 8b*. The muscle control model's feedback control gains are also shown superimposed in *Figure 8b*. These control gains were

derived for the two models by modifying the Fourier coefficients for the force and muscle activations respectively, so that the linear map from one apex to the next is best matched to that from data (*Equation 3*). While there are infinitely many controllers, even for this simple biped model, that can approximate the apex-to-apex map, our simplifying assumptions constrains the space of controllers to produce a unique fit. The leg forces and muscle activations are rectified, so that they never become negative despite feedback control (*Blum et al., 2017*). The foot placement control and leg force feedback control are activated only when the apex state deviates from nominal.

To obtain a running simulation over many steps, we break up each step into three phases: flight from apex to beginning of stance, the stance phase, and flight from the end of stance to flight apex. The control actions for the next stance are chosen at flight apex. As previously defined for the experimental data, the flight apex is when $\dot{z}$ becomes zero. In some cases, if the vertical velocity is downward when a stance phase ends ($\dot{z} < 0$), there is no flight 'apex' and the controller uses the end of stance state instead of flight apex state as input. The end of flight and thus, the beginning of stance, are determined as the moment when the distance between the body and the target foot position is exactly equal to the landing leg length. The leg length at landing is also controlled based on flight apex state, based on the linear model in *Equation 7*. At flight apex, if the distance to the next foot position is less than the target landing leg length, the runner immediately goes into stance.

Such a simulation, when started from initial conditions exactly on the nominal periodic motion, results in a perfectly periodic motion when there are no further perturbations. We then re-simulated the two point-mass running models for hundreds of steps, in the presence of noisy foot placements and leg forces or muscle activations with step-to-step variability. To model the noise in foot placement, we computed the 'desired' foot placement based on the center of mass state at flight apex (*Equations 8-9*) and then added a deviation drawn from a normal distribution, whose variance equals the foot placement variance not explained by *Equation 8*. Similarly, we incorporated imprecise control of leg forces or muscle activation in the following manner: for each stance phase, once the leg force $F(t)$ (for model-1) or muscle activation $a(t)$ (for model-2) is determined based on the center of mass state at the previous flight apex, we 'corrupt' these functions by a multiplicative noise term, so that the actual leg force or muscle activation is $F(t)(1 + \epsilon)$ or $a(t)(1 + \epsilon)$ respectively, where $\epsilon$ is drawn from a normal distribution with variance equal to the unexplained step-to-step variability in leg force magnitude. Thus, we use the unexplained variance in the foot placement and leg forces from experimental regressions as a simple model of the intrinsic noise in active control.

## Acknowledgements

We thank Andy Ruina for useful comments on an early draft. This work was supported by NSF CMMI grant 1254842 and a Schlumberger Foundation Faculty for the Future fellowship.

## Additional information

### Funding

| Funder | Grant reference number | Author |
| --- | --- | --- |
| National Science Foundation | NSF CMMI grant 1254842 | Manoj Srinivasan |
| Schlumberger Foundation | | Nidhi Seethapathi |

The funders had no role in study design, data collection and interpretation, or the decision to submit the work for publication.

### Author contributions

Nidhi Seethapathi, Conceptualization, Data curation, Software, Formal analysis, Funding acquisition, Investigation, Visualization, Methodology, Writing—original draft, Writing—review and editing; Manoj Srinivasan, Conceptualization, Data curation, Software, Supervision, Funding acquisition, Investigation, Visualization, Methodology, Project administration, Writing—review and editing

## Author ORCIDs
Nidhi Seethapathi (iD) http://orcid.org/0000-0002-5159-9717
Manoj Srinivasan (iD) http://orcid.org/0000-0002-7811-3617

## Ethics
Human subjects: The protocols were approved by the Ohio State University Institutional Review Board under protocol number 2012H0032. All subjects participated with informed consent.

## Decision letter and Author response
Decision letter https://doi.org/10.7554/eLife.38371.022
Author response https://doi.org/10.7554/eLife.38371.023

# Additional files

## Supplementary files
• Transparent reporting form
DOI: https://doi.org/10.7554/eLife.38371.018

## Data availability

All anonymized raw human running data (1GB) from this study are available at the following Dryad link: https://doi.org/10.5061/dryad.1nt24m0

The following dataset was generated:

| Author(s) | Year | Dataset title | Dataset URL | Database and Identifier |
|---|---|---|---|---|
| Seethapathi N, Srinivasan M | 2018 | Data from: Step-to-step variations in human running reveal how humans run without falling | https://doi.org/10.5061/dryad.1nt24m0 | Dryad, 10.5061/dryad.1nt24m0 |

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
