## [Decision Letter]

Thank you for submitting your article "Step-to-step variations in human running reveal how humans run without falling" for consideration by *eLife*. Your article has been reviewed by Andrew King as the Senior Editor, a Reviewing Editor, and two reviewers. The following individuals involved in review of your submission have agreed to reveal their identity: Andrew Biewener (Reviewer #1) and Greg Sawicki (Reviewer #2).

The reviewers have discussed the reviews with one another and the Reviewing Editor has drafted this decision to help you prepare a revised submission.

Summary:

How do humans stabilize themselves while running? This study of human running exploits the information contained in the natural step-to-step variability of the movement of the center of mass of the body as well as the ground reaction forces under the foot to 'reverse-engineer' the control algorithm. Using well-designed experiments and mathematical modelling, the authors provide key insights into how changes in foot placement and ground reaction force are modulated to deal with noise and perturbations to run stably. Interestingly, humans make these adjustments side-to-side within a stride, while fore-aft adjustments are made over several strides. Previous studies explained the role of elastic storage in the legs to run efficiently, the present findings shed new light on the central role of active muscle control to run stably. This finding is of broad interest to roboticists, engineers, and biomechanists interested in human and animal locomotion over terrain. The new information on how humans control their gait may be used to develop better controllers for powered exoskeletons and prostheses as well as humanoid robots.

Essential revisions:

1A) The authors tend to overstate the novelty of their findings in suggesting that past work based on a passive spring-mass running model cannot account for stable running. Past work based on spring-mass running mechanics was developed to argue for how passive-elastic properties may reduce the work of muscles needed to control leg and body movement. But these past studies do not claim that muscle work per se is unnecessary. As the author's own work (Srinivasan, 2011) and earlier papers (Alexander, 1997; Ruina et al., 2005) have noted (in addition to others – e.g. Biewener and Daley, 2007), spring-like leg behavior can result from an inelastic limb, in which muscle negative work is performed followed by equal positive work. The authors should revise to more accurately represent that understanding and approach of past workers on running dynamics in relation to their own work. We understand that some of these earlier studies may have overstated their implications towards dynamic stability during running. However, we encourage the authors to focus on their new data and model and let that speak towards the contribution of their study. Examples are listed below:

1B) The majority of those who have modeled steady running as a spring-mass system with spring-like leg dynamics do not suggest that this passive model is an accurate representation of how running is controlled, or that it would necessarily result in stable periodic motion following a perturbation. The authors' telescoping or single knee joint muscle-controlled leg model which shows net leg work (either negative or positive) is unsurprising in this regard and an expected result. Please better reflect that past workers who analyze running as spring-mass dynamics implicitly or explicitly recognize that some muscle work must be done to control the time-varying motions of a runner's CoM state for stable periodic motion.

1C) Introduction "Here, in contrast, we eschew spring-mass-like assumptions and characterize the control in terms of how humans modulate their leg force magnitude and direction." This is somewhat misleading in that spring-mass assumptions previously made by other workers were to explore the extent to which running (trotting, hopping) can be explained by passive dynamics, while recognizing that some amount of muscle work to stabilize running (or to go uphill/downhill, change direction) is necessary. The majority of prior workers do not claim that running is purely passive and does not require muscle work for control and stability. Please reflect the understanding and work in the field more accurately here.

1D) Subsection “Human-derived controller stabilizes a minimal model of bipedal running”: "Running stably cannot be purely passive and involves active leg work." Please revise so this does not overstate the significance of the authors' modeling and findings.

1E) Discussion section: The pseudo-elastic behavior of a runner's leg, which in fact could be entirely based on actuation (negative work followed by equivalent positive work) – as citations to Ruina et al., 2005 and Alexander, 1997 demonstrate, as well as the author's own past work (Srinivasan, 2011).

2) There is a large literature dedicated to understanding variability in gait (e.g., Dingwell's GEM; Chang's UCM; Hausdorff's long-term noise correlations; Laquaniti's principle components analysis etc.). These approached are predominantly based on characterizing the variability in kinematic measures as a way to assess 'stability', but they cannot say much about the underlying control and this has limited their impact on understanding control of gait mechanics more fundamentally. The approach presented here is a long-overdue, a more formal linear systems ID framework grounded in Newton-Euler mechanics. It may be worth acknowledging this and comparing/contrasting the utility of these two approaches (e.g., diagnosis vs. prediction) without overly criticizing previous approaches. E.g. by focusing one or a couple of sentences in the discussion on the key new abilities that add to previous approaches and how they could be used together in future studies.

3) The modeling approach taken here is framed in the form of a linear system ID, which is OK for small perturbations, in this case step to step variation around an average behavior. Figure 8 starts to address this in simulation, but how well does the approach hold for more explicit, larger perturbations at different phases of the gait where non-linearities might become more important? What is the plan to deal with this? Please clarify this, or the underlying limitations, in the discussion.

4) Please discuss the potential physiological mechanisms underlying the extracted controller, and suggest experiments that could be done to elucidate them if you have a new perspective on this. E.g., the authors mention a combination of feedforward and feedback processes but do not elaborate much on how vision, vestibular, proprioception, and cerebellar internal model may contribute.

5) Can this approach be applied to smaller data sets from single individuals as well? How variable is the controller across people and how much data is necessary to extract a good model?

6) Introduction "Numerous running robots have demonstrated stable periodic running, using a variety of control schemes (Raibert, 1986; Chevallereau et al., 2005; Tajima et al., 2009)." If robots are moving stably and periodically, as stated here, why then isn't human running and other animal running not stable and periodic as per the authors' lead statement (Introduction)? The earlier statement is made to argue the importance of muscle actuation for control to achieve truly stable periodic motion. Clearly both animals and robots need to do some muscle work to control movement. This inconsistency in phrasing needs to be addressed.

7) The regressions for equations 1and 2 should be shown (at least in supplemental materials).

8) Subsection “Impulse control is achieved by phase-dependent force modulations”: "compute the phase-dependent sensitivity of the GRFs to…" Exactly how are these phase-dependent sensitivities computed? (The y-axis label for these needs to be defined/explained more clearly.) This is explained in part (subsection “Linear regressions between the outputs and the inputs”) but could be made more clear and explicit when showing and referring to these patterns in Figure 4.

9) Figure 5: Swing-foot reposition and CoM state predictability of foot position – how these are determined/calculated needs to be made clear. Results are stated but their basis is not shown.

10) Figure 6: The force-velocity and force-length relations shown and implemented in the authors' model may be overly simple. F-V is an inverse hyperbolic not linear relationship and the lengthening side of F-V is much more skewed than the authors' model. (Also consider Zajac, 1989). How would a more realistic F-V relationship (inverse hyperbolic for shortening and a more steep and rapid leveling off of force in relation to lengthening) affect the authors' model?

---

## [Author Response]

Essential revisions:1A) The authors tend to overstate the novelty of their findings in suggesting that past work based on a passive spring-mass running model cannot account for stable running. Past work based on spring-mass running mechanics was developed to argue for how passive-elastic properties may reduce the work of muscles needed to control leg and body movement. But these past studies do not claim that muscle work per se is unnecessary. As the author's own work (Srinivasan, 2011) and earlier papers (Alexander, 1997; Ruina et al., 2005) have noted (in addition to others – e.g. Biewener and Daley, 2007), spring-like leg behavior can result from an inelastic limb, in which muscle negative work is performed followed by equal positive work. The authors should revise to more accurately represent that understanding and approach of past workers on running dynamics in relation to their own work. We understand that some of these earlier studies may have overstated their implications towards dynamic stability during running. However, we encourage the authors to focus on their new data and model and let that speak towards the contribution of their study. Examples are listed below:

Thank you for these constructive remarks. We do agree with the remarks by the reviewers that many authors including those that have used and pioneered the spring-mass model paradigm recognized the limitations and simplifications of spring-mass models over the years, while perhaps some authors may have taken the metaphor far as the reviewers agree. We also agree that initially spring mass models were inspired by how passive elastic properties may help reduce the muscle work needed for running and have been valuable in thinking about these tasks. So as the reviewers suggest, we have focused more on our results rather than prior spring-mass-like models. Specifically, we have made the following edits to reduce or remove our critique of spring-mass-like models and/or more explicitly attribute its weakness to prior authors (which we had perhaps done more implicitly in the prior version via parenthetical citations). We have listed many of these edits below, which are in addition to the additional ones listed in response to comments 1B, 1C, 1D, and 1E.

Original: “The classic modeling paradigm for running control assumes that the human leg behaves like a linear spring (Blickhan, 1989; McMahon and Cheng, 1990; Blickhan and Full, 1993).”

Revised: “One classic modeling paradigm for running control assumes that the human leg behaves like a linear spring (Blickhan, 1989; McMahon and Cheng, 1990; Blickhan and Full, 1993). This paradigm has been used to argue how passive-elastic properties may reduce muscle work needed for locomotion (Alexander and Vernon, 1975; Alexander, 1990) and has been useful in examining different aspects of locomotion in a simplified setting.”

Original: “These models are usually based on fitting to the average periodic center of mass motion during 38 running (Blickhan and Full, 1993; Geyer et al., 2006; Srinivasan and Holmes, 2008). However, understanding running stability requires understanding how deviations from the average motion 40 are controlled.”

Revised: “These models have been successful in fitting the average periodic center of mass motion during running (Blickhan and Full, 1993; Geyer et al., 2006; Srinivasan and Holmes, 2008). Understanding running stability requires understanding how deviations from the average motion are controlled.”

Original: “Spring-like leg mechanics cannot explain how deviations from the average motion are controlled (Maus et al., 2015).”

Revised: “It has been previously recognized that spring-like leg mechanics cannot explain how deviations from the average motion are controlled and are eventually attenuated (e.g., Ghigliazza et al., 2005; Biewener and Daley, 2007; Maus et al., 2015).”

Original: “Here, we use more general models of human locomotion, rooted in Newtonian mechanics, to implement running control, without making any spring-like parametric assumptions (Srinivasan 2011).”

Revised: “Here, we examine the role of active muscle control in running stability, using more general models of human locomotion rooted in Newtonian mechanics (Srinivasan, 2011).”

Edited version: In the following sentence, we have removed the reference to spring-mass model.

“We uncover how such center of mass control is achieved. We then implement this human-derived controller on a simple mathematical model of a biped (Srinivasan, 2011), showing that this biped model runs without falling down, despite incessant noise-like perturbations, large external perturbations, and on uneven terrain.”

Revised: “Here, we have used a simpler `non-parametric' model to directly describe the control of stance leg force or activation (Figure 8).”

1B) The majority of those who have modeled steady running as a spring-mass system with spring-like leg dynamics do not suggest that this passive model is an accurate representation of how running is controlled, or that it would necessarily result in stable periodic motion following a perturbation. The authors' telescoping or single knee joint muscle-controlled leg model which shows net leg work (either negative or positive) is unsurprising in this regard and an expected result. Please better reflect that past workers who analyze running as spring-mass dynamics implicitly or explicitly recognize that some muscle work must be done to control the time-varying motions of a runner's CoM state for stable periodic motion.

Thank you for these remarks. In our previous writing of the paragraph, we weren’t clear enough that we were attributing some of these reasoning to prior researchers, although we had cited them parenthetically. We have re-written the relevant paragraph to hopefully reflect this prior understanding better:

“It is expected that any running controller that achieves asymptotic stability will need to perform net mechanical work in response to perturbations that decrease or increase the body’s mechanical energy (Ghigliazza et al., 2005; Srinivasan and Holmes, 2008; Maus et al., 2015). Our results are consistent with such expectation, as illustrated by the work-loops with net mechanical work in Figure 11. Energy-conservative spring-like leg behavior does not allow such net mechanical work and can achieve only partial asymptotic stability, not being able to handle energy-changing perturbations (as noted by Ghigliazza et al. (2005)).”

1C) Introduction "Here, in contrast, we eschew spring-mass-like assumptions and characterize the control in terms of how humans modulate their leg force magnitude and direction." This is somewhat misleading in that spring-mass assumptions previously made by other workers were to explore the extent to which running (trotting, hopping) can be explained by passive dynamics, while recognizing that some amount of muscle work to stabilize running (or to go uphill/downhill, change direction) is necessary. The majority of prior workers do not claim that running is purely passive and does not require muscle work for control and stability. Please reflect the understanding and work in the field more accurately here.

Thank you for these remarks. We have now edited the following sentence:

“Here, in contrast, we eschew spring-mass-like assumptions and characterize the control in terms of how humans modulate their leg force magnitude and direction.” to “Here, we directly characterize the control in terms of how humans modulate their leg force magnitude and direction during running.”

1D) Subsection “Human-derived controller stabilizes a minimal model of bipedal running”: "Running stably cannot be purely passive and involves active leg work." Please revise so this does not overstate the significance of the authors' modeling and findings.

Thank you. As suggested, we have edited the original subsection “Running stably cannot be purely passive and involves active leg work” to be more ‘illustrative’ of prior expectations, more explicitly acknowledging various authors on the topic.

“Non-zero leg work for energy-changing perturbations.

Figure 11 illustrates the leg work-loop for the unperturbed run (net zero work) and when positive perturbations are applied to sideways and fore-aft velocities, and vertical positions. All such positive perturbations result in net negative work on the first step after the perturbation, reflected in the work-loops with net negative area within them. Such net positive or negative leg work is clearly necessary to recover from perturbations that change the total mechanical energy of the runner, as was recognized in prior discussions of the energy-neutral spring-mass model of running (Ghigliazza et al., 2005; Biewener and Daley, 2007; Srinivasan and Holmes, 2008).”

1E) Discussion section: The pseudo-elastic behavior of a runner's leg, which in fact could be entirely based on actuation (negative work followed by equivalent positive work) – as citations to Ruina et al., 2005 and Alexander, 1997 demonstrate, as well as the author's own past work (Srinivasan, 2011).

Thank you for these remarks. We have now deleted the sentence that previously was” Our results suggest…”, as described in response to reviewer comment 1B.

Regarding the seventh paragraph of the Discussion section, we assume that the reviewers are agreeing with us and simply drawing our attention to our paragraph regarding prior discussions of pseudo-elastic leg behavior in running. We have left this paragraph mostly intact, editing it down slightly as follows:

“Indeed, it is generally thought that even the spring-mass-like steady state center of mass motion in running is due to considerable muscle action, and has been termed pseudo-elastic (Ruina et al., 2005) or pseudo-compliant (Alexander, 1997). Remarkably, energy-optimal running movements in models with no leg springs produce similar spring-mass-like center of mass trajectories (Srinivasan, 2011), with leg muscles performing equal amounts of positive and negative work.”

2) There is a large literature dedicated to understanding variability in gait (e.g., Dingwell's GEM; Chang's UCM; Hausdorff's long-term noise correlations; Laquaniti's principle components analysis etc.). These approached are predominantly based on characterizing the variability in kinematic measures as a way to assess 'stability', but they cannot say much about the underlying control and this has limited their impact on understanding control of gait mechanics more fundamentally. The approach presented here is a long-overdue, a more formal linear systems ID framework grounded in Newton-Euler mechanics. It may be worth acknowledging this and comparing/contrasting the utility of these two approaches (e.g., diagnosis vs. prediction) without overly criticizing previous approaches. E.g. by focusing one or a couple of sentences in the discussion on the key new abilities that add to previous approaches and how they could be used together in future studies.

Thank you for these remarks. We have now added a new paragraph that show how our methods can complement these other techniques:

“Some past work on inferring stability from variability focused on kinematic measures of variability such as Floquet multipliers, finite-time Lyapunov exponents (Dingwell et al., 2001) and long-term correlations in walking and running variability (Hausdorff et al., 1995; Jordan et al., 2006; Kaipust et al., 2012). Such measures can provide discriminative diagnostic measures (Kaipust et al., 2012), but do not attempt to provide a causal narrative about how locomotion is controlled. Our approach here, rooted in Newton-Euler mechanics, is able to discover potential causal strategies underlying locomotion stability, and by extension, could inform treatment of pathological unstable movements in addition to diagnosis. Other past studies has used variants of the principal component analysis (Cappellini et al., 2006; Maus et al., 2015) to demonstrate that the intrinsic variability in human locomotion may reside in a lower dimensional manifold (Cappellini et al., 2006; Maus et al., 2015, Dingwell et al., 2010; Chang et al., 2009; Yen et al., 2009; Maus et al., 2015). Here, we used a physics-based dimensionality reduction to examine the dominant control strategies, by focusing on how the center of mass is controlled through forces.”

3) The modeling approach taken here is framed in the form of a linear system ID, which is OK for small perturbations, in this case step to step variation around an average behavior. Figure 8 starts to address this in simulation, but how well does the approach hold for more explicit, larger perturbations at different phases of the gait where non-linearities might become more important? What is the plan to deal with this? Please clarify this, or the underlying limitations, in the discussion.

Thank you for these remarks. We have now added the following remarks in the Discussion section:

“We have focused on linear relations between state deviations and control as it is naturally suited for small deviations and perturbations that our data contains. In future work, we hope to examine the range of perturbation sizes for which this linear description is accurate by comparing this linear control to responses to experiments with larger perturbations, also inferring nonlinear descriptions should they improve predictive capability.”

4) Please discuss the potential physiological mechanisms underlying the extracted controller, and suggest experiments that could be done to elucidate them if you have a new perspective on this. E.g., the authors mention a combination of feedforward and feedback processes but do not elaborate much on how vision, vestibular, proprioception, and cerebellar internal model may contribute.

Thank you for these remarks. We have now added the following paragraph in the Discussion section, addressing the sensory integration that may be needed to accomplish feedback control:

“Center of mass state or other body state information needed for feedback control could be estimated by the nervous system by integrated sensory signals from vision (Patia, 1997), proprioceptive sensors (especially when the foot is on the ground (Sainburg et al., 1995)), and vestibular sensors (Angelaki and Cullen, 2008), potentially in combination with predictive internal models (Walpert et al., 1995; Cullen, 2004). In future work, repeating the calculations herein (for instance, Figure 5) for experiments that systematically block or degrade (say, by adding sensory nose) one or more of these sensors may well tell us the contributions of theses sensors to running control. We speculate that the most available relevant sensory information is used, perhaps analogous to an optimal state estimator (Kuo, 2005; Srinivasan, 2009) and degrading one sensor may result in sensory re-weighting on a slow time-scale (Carver et al., 2006; Assländer and Peterka, 2014).”

5) Can this approach be applied to smaller data sets from single individuals as well? How variable is the controller across people and how much data is necessary to extract a good model?

Thank you for this question. We have now added a completely new illustrative figure (Figure 12) that provides information on these questions. We have also added the following paragraph in the Discussion section, remarking on these issues.

“The results we have presented have been for data pooled over all subjects. Performing the regressions for data from individual subjects indicates that the dominant terms in the inferred controllers are similar for all subjects; the subject to subject variability in the estimated control gains are shown in Figure 12a. Figure 12b illustrates how the accuracy of an estimated control gain depends on the number of strides used for regression. For such linear regressions, the error estimate (standard deviation) is expected to decrease with *N*_stride_ like 1∕*N*^2^_stride_, so that a factor of 10 decrease in error requires a 100-fold increase in sample size (Wang and Srinivasan, 2012; Hamilton, 1994). This dependence on *N*_stride_ may guide selection of sample sizes for future experimental designs.”

6) Introduction "Numerous running robots have demonstrated stable periodic running, using a variety of control schemes (Raibert, 1986; Chevallereau et al., 2005; Tajima et al., 2009)." If robots are moving stably and periodically, as stated here, why then isn't human running and other animal running not stable and periodic as per the authors' lead statement (Introduction)? The earlier statement is made to argue the importance of muscle actuation for control to achieve truly stable periodic motion. Clearly both animals and robots need to do some muscle work to control movement. This inconsistency in phrasing needs to be addressed.

Thank you for pointing to the inconsistency in the wording. We have now edited as follows, which hopefully removes the inconsistency.

"Some running robots have demonstrated stable running (that is, running without falling down) using a variety of control schemes (Raibert, 1986; Chevallereau et al., 2005; Tajima et al., 2009; Nelson et al., 2019)."

That is, we have edited the sentence by (1) removing the word ‘periodic’ when referring to robot running and (2) also clarified what we mean by ‘stable robot running’, that is, ‘running without falling down.’ By stable running, we don’t mean that the motion is perfectly periodic, but just that motion does not result in a fall. We hope our edits has addressed the perceived inconsistency between this sentence and the earlier sentence about human running not being perfectly periodic.

7) The regressions for equations 1and 2 should be shown (at least in supplemental materials).

Thank you for this suggestion. We have now included a new Figure 3 in the article that shows this regression, included below with its caption.

“of the stride cycle, specifically, the time fraction *Ø*_stance_ of stance (Figure 2b). To explain this phase-dependent force variability within a single step, we compute the phase dependent sensitivity of (*F_x_, Fy, F_z_*) to the center of mass state as follows. For each output, say *F_x_*, we divide the stance duration into 20 phases and compute a linear model for *F_x_*at each of those phases, all with (*ẋ_a_, ẏ_a_, z_a_*) as inputs. We refer to the coefficients in these linear models as a function of the phase *Ø*_stance_ as the phase-dependent sensitivities of the GRFs (Figure 5) to the corresponding predictor variable in (*ẋ_a_, ẏ_a_, z_a_*).”

8) Subsection “Impulse control is achieved by phase-dependent force modulations”: "compute the phase-dependent sensitivity of the GRFs to…" Exactly how are these phase-dependent sensitivities computed? (The y-axis label for these needs to be defined/explained more clearly.) This is explained in part (subsection “Linear regressions between the outputs and the inputs”) but could be made more clear and explicit when showing and referring to these patterns in Figure 4.9) Figure 5: Swing-foot reposition and CoM state predictability of foot position – how these are determined/calculated needs to be made clear. Results are stated but their basis is not shown.

Thank you for these remarks. We have now added the following additional paragraph in the Materials and methods section.

“In addition to the regressions described above using the flight apex state as the predictor, we used the center of mass state (Δ*ẋ_a_,* Δ*ẏ_a_,* Δ*z_a_*) at different phases over the previous step to predict each of the stance phase outputs. Specifically, for each stance phase output, we performed 20 regressions, each using the center of mass state at one of 20 equally spaced gait phases over the previous step, where one full step is defined as starting and ending at a touch-down. This analysis allows us to investigate the predictive ability of the center of mass state at different phases. For these phase-dependent regressions, in addition to using the center of mass state as the predictor, we repeated the calculations using the swing foot state (position and velocity relative to the center of mass), so as to compare the different predictive abilities as in Figure 6.”

We have also expanded the figure caption from

**“**Figure 6. Swing foot control before foot placement.The fraction of sideways foot placement (panel a) and fore-aft foot placement (panel b) variance at beginning of stance predicted by the center of mass (CoM) state or swing foot state during the previous one step (flight and stance). To produce this figure, a sequence of linear models were built for predicting the foot placement based on the center of mass state or swing foot state during different phases through the previous step. We plot the *R*2 value corresponding to these linear models (that is, fraction of variance explained) as a function of the gait phase used for the prediction; the gait phase is represented as the fraction of a step starting from beginning of previous stance phase. The solid and dashed lines represent right and left foot placements respectively.”

10) Figure 6: The force-velocity and force-length relations shown and implemented in the authors' model may be overly simple. F-V is an inverse hyperbolic not linear relationship and the lengthening side of F-V is much more skewed than the authors' model. (Also consider Zajac, 1989). How would a more realistic F-V relationship (inverse hyperbolic for shortening and a more steep and rapid leveling off of force in relation to lengthening) affect the authors' model?

Thank you for these remarks. We have now added the following sentences so as to comment on this model simplification:

“Further, we've made simplifying assumptions about muscle architecture, muscle properties (linear force-velocity relations), and muscle activation, which are meant to capture the primary qualitative dynamical features of muscles, rather than model them quantitatively precisely. For instance, the linear force-velocity relation may be sufficient to produce damping-like muscle behavior when activated, but this damping behavior may be accentuated by a nonlinear forcevelocity relation. Future work will also involve obtaining controllers for more complex biped models and muscle models.”